# Glycolytic disruption restricts *Drosophila melanogaster* larval growth via the cytokine Upd3

**Madhulika Rai**[1], **Hongde Li**[1], **Robert A. Policastro**[1], **Robert Pepin**[2],
**Gabriel E. Zentner**[1†], **Travis Nemkov**[3], **Angelo D'Alessandro**[3], **Jason M. Tennessen**[1,4*]

**1** Department of Biology, Indiana University, Bloomington, Indiana, United States of America,
**2** Department of Chemistry, Indiana University, Bloomington, Indiana, United States of America,
**3** Department of Biochemistry and Molecular Genetics, University of Colorado Anschutz Medical Campus, Aurora, Colorado, United States of America, **4** Member, Melvin and Bren Simon Cancer Center, Indianapolis, Indiana, United States of America

† Deceased.
* jtenness@iu.edu

## Abstract

*Drosophila* larval growth requires efficient conversion of dietary nutrients into biomass. Lactate dehydrogenase (Ldh) and glycerol-3-phosphate dehydrogenase (Gpdh1) support this larval metabolic program by cooperatively promoting glycolytic flux. Consistent with their cooperative functions, the loss of both enzymes, but not either single enzyme alone, induces a developmental arrest. However, Ldh and Gpdh1 exhibit complex and often mutually exclusive expression patterns, suggesting that the lethal phenotypes exhibited by *Gpdh1; Ldh* double mutants could be mediated non-autonomously. Supporting this possibility, we find that the developmental arrest displayed by double mutants extends beyond simple metabolic disruption and instead stems, in part, from changes in systemic growth factor signaling. Specifically, we demonstrate that the simultaneous loss of Gpdh1 and Ldh results in elevated expression of Upd3, a cytokine involved in Jak/Stat signaling. Furthermore, we show that *upd3* loss-of-function mutations suppress the *Gpdh1; Ldh* larval arrest phenotype, indicating that Upd3 signaling restricts larval development in response to decreased glycolytic flux. Together, our findings reveal a mechanism by which metabolic disruptions can modulate systemic growth factor signaling.

## Author summary

Animal development requires precise coordination between growth, metabolism, and nutrition. When animals are raised under ideal conditions, dietary nutrients support optimal growth by driving both biosynthesis and energy production. In contrast, environmental stressors such as starvation often necessitate a metabolic shift towards using limited nutrients for survival. However, the coordination between growth and metabolism under stressful conditions is complex, as

**Data availability statement:** The RNA-seq data discussed in this publication have been deposited in NCBI's Gene Expression Omnibus and are accessible through GEO Series accession number GSE119334 (https://www.ncbi.nlm.nih.gov/geo/query/acc.cgi?acc=GSE119334) and GSE121876 (https://www.ncbi.nlm.nih.gov/geo/query/acc.cgi?acc=GSE121876).

**Funding:** J.M.T. is supported by the National Institute of General Medical Sciences of the National Institutes of Health under a R35 Maximizing Investigators' Research Award 1R35GM119557 (https://www.nigms.nih.gov). The funders had no role in study design, data collection and analysis, decision to publish, or preparation of the manuscript. The salaries of Madhulika Rai, Hongde Li, and Jason M. Tennessen were supported with funds from NIGMS/NIH R35 MIRA award 1R35GM119557.

**Competing interests:** The authors have declared that no competing interests exist.

there is a need to monitor and coordinate metabolic flux across all tissues of the developing animal. When a single tissue encounters metabolic limitations, peripheral tissues must adjust their growth rates to prevent asynchronous development. In this study, we use the fruit fly *Drosophila melanogaster* to understand how metabolic insults in the muscle lead to changes in the growth rate of other tissues. Our findings reveal that juvenile muscle releases a signal in response to disrupted sugar metabolism. This signal activates growth arrest in tissues throughout the body and inhibits the production of a key steroid hormone that promotes developmental progression. In this manner, a metabolic disruption in a single tissue, in this case the muscle, can affect the growth rate of the entire animal.

## Introduction

Animal development requires the precise integration of environmental and metabolic cues with intrinsic growth signaling pathways [1,2]. Under ideal growth conditions, the metabolic pathways involved in central carbon metabolism efficiently convert dietary nutrients into the biomass and energy that are required to support rapid growth. Conversely, environmental stressors such as starvation, infection, and toxicant exposure necessitate metabolic reprogramming to maintain survival until growth conditions improve [3–5]. The coordination between growth and metabolism under stressful environmental conditions, however, is complex, as there is a need to monitor and coordinate metabolic flux across all tissues of the developing animal [6]. When a single tissue encounters metabolic limitations, peripheral tissues must adjust their growth rates to prevent asynchronous development. This topic of systemic metabolic communication has become a burgeoning area of research, with numerous studies examining how nutrient-sensing proteins and secreted signaling molecules coordinate growth across multiple tissues [7–9]. Nevertheless, the specific metabolic signals that trigger growth signaling across organ systems are only beginning to emerge [10–12].

The fruit fly, *Drosophila melanogaster,* is a powerful system for exploring how metabolism is coordinated with developmental growth [13]. The ~200-fold increase in body mass that occurs during the four days of *Drosophila* larval development represents an appealing model for exploring metabolic mechanisms that convert nutrients into biomass and energy [14]. In this regard, previous studies revealed that this impressive growth rate is supported by a coordinated increase in the expression of genes involved in glycolysis, the pentose phosphate pathway, and related aspects of carbohydrate metabolism [15–17]. The resulting larval metabolic program exhibits hallmark features of aerobic glycolysis – a specialized form of carbohydrate metabolism that is ideally suited for rapid biomass production. Considering that several types of cancer cells also rely on aerobic glycolysis for growth and survival [18], *Drosophila* larvae present a compelling model to study this biosynthetic state.

We previously demonstrated that the larval metabolic program requires the enzymes Ldh and Gpdh1, which cooperatively regulate NAD+/NADH redox balance and promote glycolytic flux [19]. Intriguingly, while the loss of either single enzyme has minimal effect on overall larval growth, *Gpdh1; Ldh* double mutants experience a larval developmental arrest, suggesting functional redundancy, where each enzyme partially compensates for the loss of the other [19]. However, neither Ldh nor Gpdh1 are ubiquitously expressed in larvae [15,19,20], raising questions as to how these enzymes serve compensatory roles if not expressed in a strictly overlapping pattern. Here we address this question by reexamining the expression pattern of these enzymes as well as the *Gpdh1; Ldh* double mutant phenotypes.

Our analysis of Gpdh1 and Ldh expression during larval development confirms previous observations that these enzymes are expressed in a complex and often mutually exclusive expression pattern [15,20]. Moreover, we demonstrate that the loss of both enzymes within larval muscle can inhibit larval growth, thus hinting at a link between glycolytic disruption and systemic growth signaling. Consistent with these observations, RNA-seq analysis of *Gpdh1; Ldh* double mutants reveals altered expression of several secreted signaling molecules, including increased expression of the cytokine Upd3, which was previously shown to regulate larval growth in response to cellular stress [21]. Using a genetic approach, we find that *upd3; Gpdh1; Ldh* triple mutants can survive larval development and eventually eclose into adults, indicating that increased Upd3 expression plays a role in the larval arrest phenotype displayed by *Gpdh1; Ldh* double mutants. Furthermore, we demonstrate that the steroid hormone 20-hydroxyecdysone (20E) also plays a role in this phenotype, as a dietary supplement of 20E allows *Gpdh1; Ldh* double mutants to complete larval development. Overall, our findings reveal that the developmental delay and lethal phenotype associated with *Gpdh1; Ldh* double mutants are not simply the result of metabolic failure but rather stem, in part, from changes in systemic growth signaling.

## Results

### Ldh and Gpdh1 influence larval tissue growth in a nonautonomous manner

To better understand how Ldh and Gpdh1 cooperatively promote *Drosophila* larval growth, we examined the spatial expression pattern of both enzymes. Our analysis revealed that Ldh and Gpdh1 are often expressed in a mutually exclusive manner. In the Malpighian tubules, for example, *Ldh* is highly expressed in the stellate cells while *Gpdh1* is not (Fig 1A–1A'''). Similarly, cells of the larval midgut tend to express high levels of either Ldh or Gpdh1, but we rarely observed a cell expressing detectable levels of both enzymes (Figs 1B–1B''' and S1A–S1D). This trend is also apparent in the central nervous system and prothoracic gland (PG), where although both *Ldh* and *Gpdh1* are co-expressed in many cells of the CNS, *Gpdh1*, but not *Ldh*, is expressed at detectable levels in neural stem cells and the PG (Figs 1C–1D''' and S1E–S1H). Finally, both the fat body and salivary glands express relatively high levels of Gpdh1 while Ldh is mostly undetectable in these tissues (Figs 1A–1A''' and S1I–S1P), a result that is supported by previous observations [15,20]. In fact, the only tissue where we observed uniform and overlapping expression of both enzymes was in the larval body wall muscle (Fig 1E–1E''').

The complex *Ldh* and *Gpdh1* expression patterns raise the question of how these enzymes cooperatively regulate larval metabolism. One explanation is that the loss of one enzyme alters the spatial expression pattern of the other; however, this hypothesis is not supported by our observations. For example, Ldh expression is nearly undetectable in wild-type salivary glands and is not increased in *Gpdh1^{A10/B18}* mutant salivary glands (S2A–S2G Fig). Similarly, salivary gland Gpdh1 expression is not further elevated in *Ldh^{16/17}* mutants (S2H–S2N Fig). We also observe no increase in Ldh expression within the fat body of *Gpdh1^{A10/B18}* mutants (S3A–S3G Fig), although we do see a slight increase in Gpdh1 expression within *Ldh^{16/17}* mutant fat bodies (S3H–S3N Fig). These results are also consistent with a previous observation that Gpdh1 levels are not increased in *Ldh^{16/17}* mutant clones within the larval CNS [19].

Our findings suggest that the *Gpdh1^{A10/B18}; Ldh^{16/17}* double mutant growth defects are not simply the combined result of metabolic dysfunction within individual cells but rather stem from changes in systemic growth signaling. As a first step towards testing this hypothesis, we used a previously described *UAS-Ldh-RNAi* transgene to deplete *Ldh* expression in

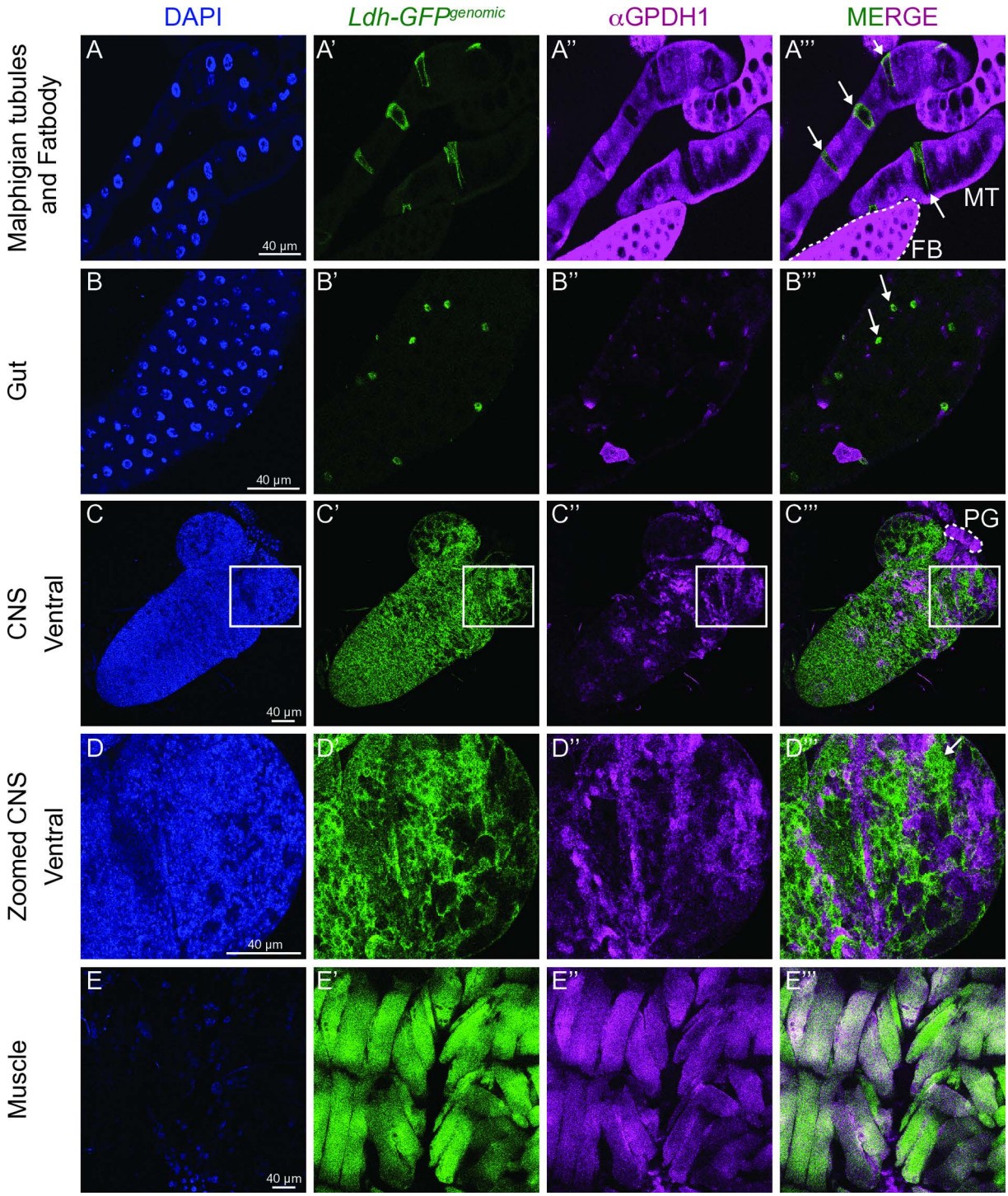

**Fig 1. _Ldh_ and _Gpdh1_ expression patterns are complex and non-strictly overlapping.** Representative confocal images of second instar larval tissues expressing _Ldh-GFP^Genomic_ and immuno-stained with αGpdh1 antibody. DAPI is shown in blue, Ldh-GFP and Gpdh1 are represented in green and magenta, respectively. The rightmost panel displays the merged images of Ldh-GFP and Gpdh1 staining. (A-A''') Malpighian tubules, (B-B''') gut, (C-C''') ventral side of CNS, (D-D''') magnification of the outlined region of interest in (C-C'''), and (E-E''') muscles. Arrows in (A''), (B''') and (D''') denote non-overlapping Ldh expression. The scale bar represents 40 μM.

the muscle (*Mef2R-GAL4*) of *Gpdh1^A10/B18^* mutant [22]. If *Gpdh1^A10/B18^; Ldh^16/17^* double mutants arrest simply due to metabolic dysfunction, we would expect that tissues unaffected by *Ldh-RNAi* would continue to grow at a normal rate. In contrast, if loss of both *Gpdh1* and *Ldh* induces changes in systemic growth signaling, *Ldh-RNAi* within the muscles of *Gpdh1* mutants should induce a global developmental delay. Our findings support the latter model, as *Gpdh1* mutants expressing *Mef2R-Ldh-RNAi* displayed significantly reduced larval growth and slower overall development, as determined by time to pupation (Fig 2A–2C). Moreover, these developmental delays were apparent in tissues not expressing the *Ldh-RNAi* transgene. For example, the salivary glands of age-matched *Gpdh1; Mef2R-Ldh-RNAi* larvae were significantly smaller than those of either the wild-type control, *Gpdh1* mutant, or *Mef2R-Ldh-RNAi* animals (Fig 2D–2G). Note that the

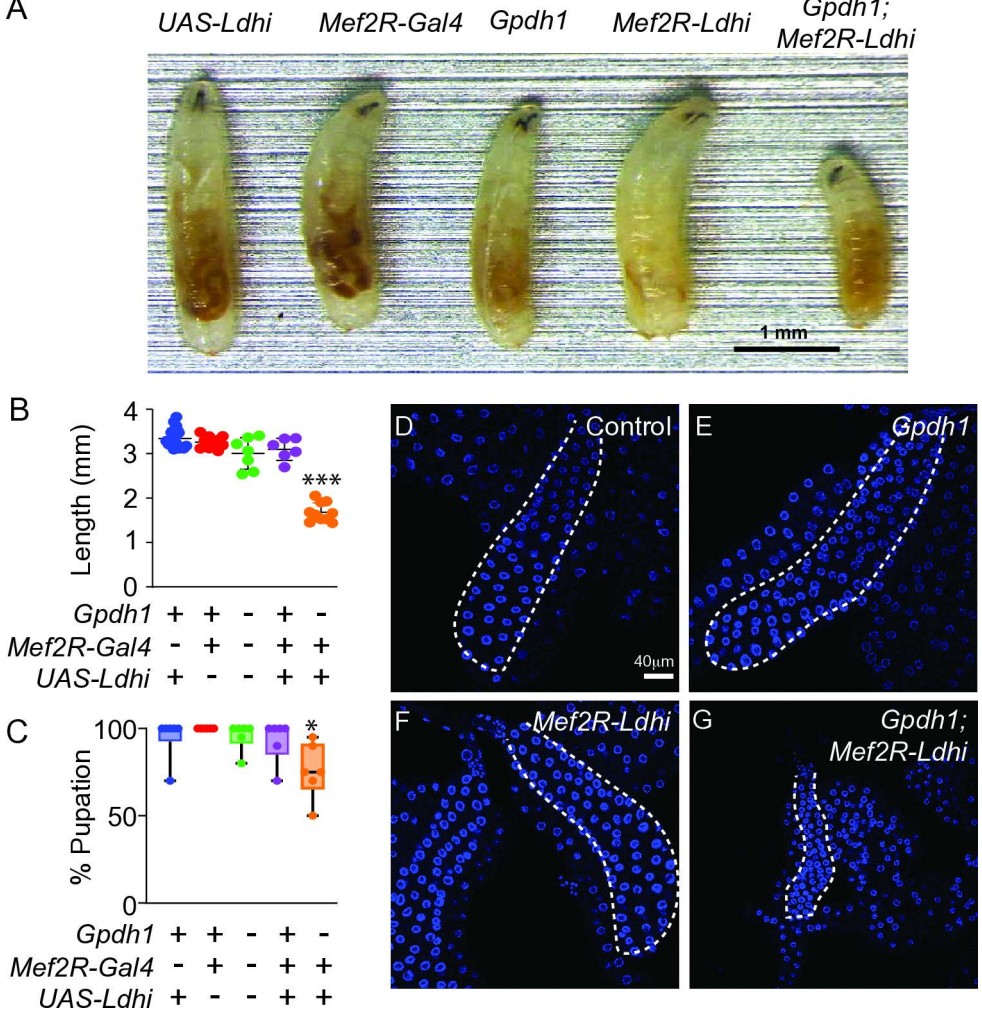

**Fig 2. Tissue-specific loss of Ldh and Gpdh1 induces systemic growth defects.** Growth and development of both control (*Mef2R-Gal4* and *UAS-Ldhi*) and mutant strains (*Gpdh1^A10/B18^* mutants, *Mef2R-Ldhi*, and *Gpdh1^A10/B18^; Mef2R-Ldhi*) were monitored throughout larval development. Note that *UAS-Ldhi* is an abbreviation for *UAS-Ldh-RNAi*. (A) Representative larval images from the indicated genotypes at 6 days after egg-laying (AEL). The scale bar represents 1 mm. (B-C) Quantification of (B) larval length and (C) percent of larvae that pupated eight days AEL from the indicated genotypes. All experiments were repeated a minimum of three times. (B, C) n ≥ 5 biological replicates. Data presented as a scatter plot with the lines representing the mean and standard deviation. *P*-values were calculated using an ANOVA followed by a Holm-Sidak test. *$P < 0.05$, ***$P < 0.001$. (D-G) Representative images of salivary glands dissected from the indicated genotypes. DAPI is shown in blue. The scale bar represents 40 μM. The scale bar in (D) applies to (E,F,G).

*Mef2R-GAL4* driver does not drive expression in salivary glands at this stage of larval development (S4 Fig). We would also note that our attempts to conduct the reciprocal experiment were unsuccessful because currently available *Gpdh1-RNAi* lines do not reduce Gpdh1 protein levels in muscle (S5 Fig). Regardless, these results suggest that loss of both enzymes within a single tissue can globally influence larval growth and development.

## Upd3 impedes larval growth in *Gpdh1; Ldh* double mutants

To better understand how the simultaneous loss of Ldh and Gpdh1 disrupts larval development, we used RNA-seq to analyze gene expression in *Ldh[16/17]* mutants, *Gpdh1[A10/B18]* mutants, and *Gpdh1[A10/B18]; Ldh[16/17]* double mutants relative to genetically matched heterozygous controls (S1 Table). While our approach identified several genes that were significantly changed in both single mutants (Figs 3A, 3B and S6), for the purpose of this analysis we focused on those genes that were only altered in double mutant larvae (S2 Table). Among the genes that were significantly down- or up-regulated in *Gpdh1[A10/B18]; Ldh[16/17]* double mutants but not the single mutants, we identified four secreted factors that were previously shown to systemically regulate larval growth and metabolism: *peptidoglycan recognition protein SC2* (*PGRP-SC2*; FBgn0043575), *peptidoglycan recognition protein LA*, (*PGRP-LA*; FBgn0035975), *dawdle* (*daw*; FBgn0031461), and *unpaired 3* (*upd3*; FBgn0053542) [8,21,23]. While each of these molecules is known to function in interorgan communication and could contribute to the *Gpdh1[A10/B18]; Ldh[16/17]* double mutant phenotype, we decided to focus on the cytokine Upd3 for the remainder of this study because recent studies have shown that Upd3 inhibits larval development in response to both environmental cues and intracellular stress [8,21,24].

In order to confirm that Upd3 expression is increased in the *Gpdh1[A10/B18]; Ldh[16/17]* double mutant arrest phenotype, we examined *upd3* expression in larval body wall muscle. Our decision to focus on larval muscle stems from our observations that (i) both Gpdh1 and Ldh are coexpressed in this tissue and (ii) that *Mef2R-Ldh-RNAi* induced a developmental arrest in *Gpdh1* mutants. Using a previously described *upd3-Gal4, UAS-GFP* (*upd3-GFP*) reporter [25], we observed a significant increase in *upd3-GFP* expression in the muscles of *Gpdh1[A10/B18]; Ldh[16/17]* double mutants relative to single mutant controls (Fig 3C–3G). We would also highlight the fact that the muscles of *Gpdh1* and *Ldh* single mutants exhibited increased *upd3-GFP* expression when compared to the heterozygous controls, albeit not to the same levels as the double mutants, suggesting that *upd3* expression within the muscle is sensitive to metabolic changes that are held in common between the two single mutants (Fig 3C–3G).

As a complement to the *upd3-GFP* analysis described above, we also examined JNK signaling within the muscle. Since JNK signaling is well documented to activate *upd3* expression in response to cellular stress [21,26–31], we would anticipate that the larval muscle of *Gpdh1[A10/B18]; Ldh[16/17]* double mutant would also display increased JNK signaling. Indeed, using a polyclonal antibody that specifically recognizes the activated and phosphorylated form of JNK [31,32], we found that double mutant larvae exhibited elevated levels of active JNK in the muscle when compared with either the heterozygous control or the single mutant controls (Figs 4A–4L and S7).

The gene expression studies described above support a model in which loss of Gpdh1 and Ldh activity result in elevated JNK signaling and Upd3 expression. To test the possibility that Upd3 from the muscle could contribute to the double mutant larval arrest phenotype, we first expressed a *UAS-upd3* transgene in muscle using *Mef2R-Gal4*. The resulting larvae were smaller than age-matched controls (Fig 5A and 5B), and more than half of the animals failed to enter metamorphosis (Fig 5C and 5D). Intriguingly, *Mef2R-upd3* expression also induced abnormal pupal development, with *Mef2R-upd3* pupae displaying a hybrid of pupal and larval characteristics (Fig 5C and 5D). In most instances, the anterior portion of these animals formed pupal cases while the posterior retained a larval cuticle (Fig 5C). This is an intriguing observation considering that elevated Upd3 signaling was previously demonstrated to disrupt 20E signaling [21], and suggests that Upd3 can inhibit the larval-to-pupal transition.

As a second approach to determine if increased Upd3 expression within *Gpdh1[A10/B18]; Ldh[16/17]* double mutant impedes larval development, we generated *upd3[Δ]; Gpdh1[A10/B18]; Ldh[16/17]* triple mutants, with the hypothesis that loss of Upd3

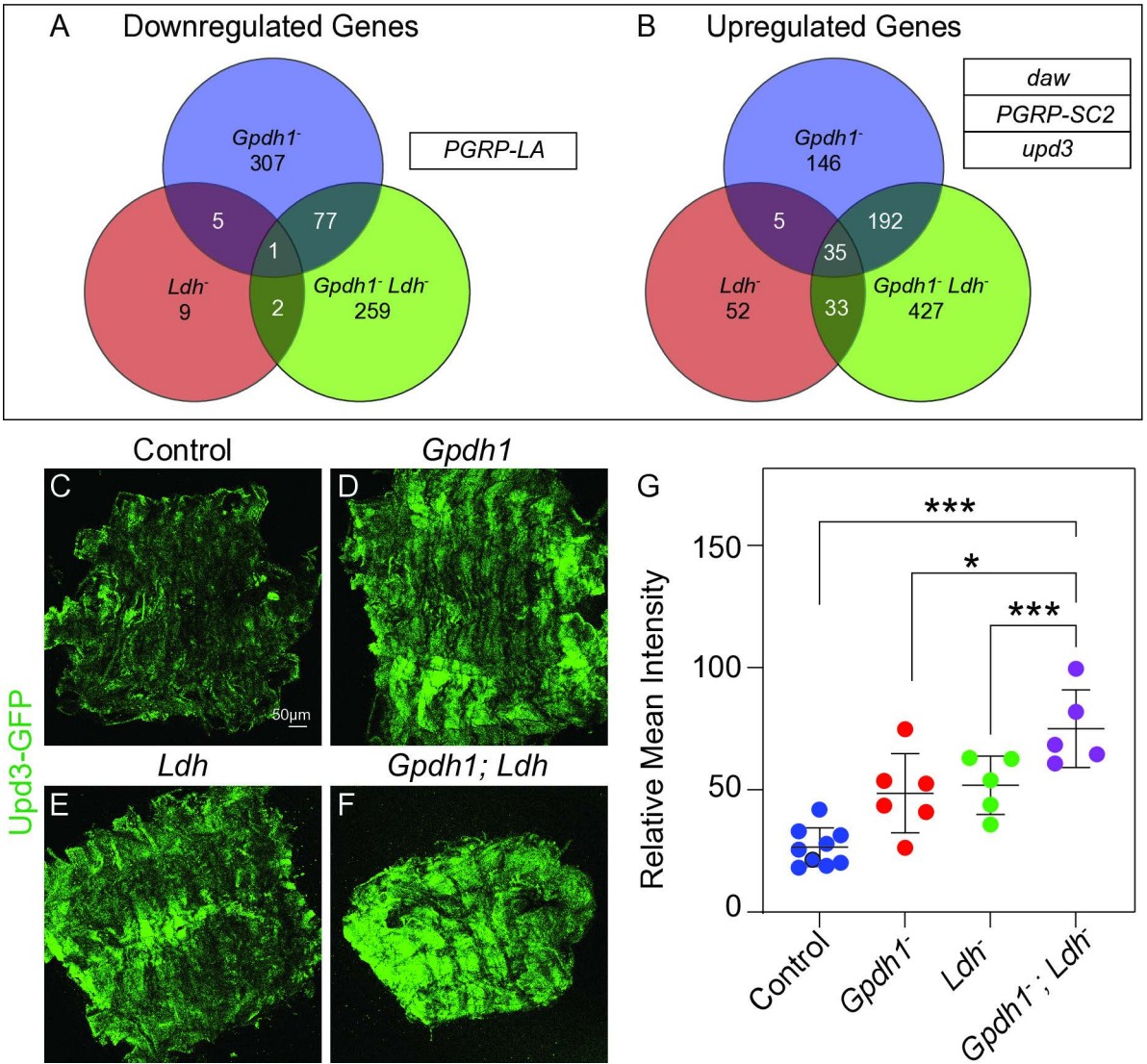

**Fig 3. Gpdh1; Ldh double mutants display increased upd3 expression.** RNA-seq was used to analyze gene expression in L2 larvae of $Gpdh1^{A10/B18}$ mutants relative to $Gpdh1^{A10/+}$ heterozygous controls, $Ldh^{16/17}$ mutants relative to $Ldh^{16/+}$ heterozygous controls, and $Gpdh1^{A10/B18}; Ldh^{16/17}$ double mutants relative to $Gpdh1^{A10/+}; Ldh^{16/+}$ heterozygous controls. (A-B) Venn diagrams showing the overlap between the number of genes that were either (A) down-regulated or (B) upregulated in either single mutant ($Gpdh1^{A10/B18}$ and $Ldh^{16/17}$) as well as the double mutant ($Gpdh1^{A10/B18}; Ldh^{16/17}$) relative to the respective heterozygous control strains. Genes listed in (A, B) encode secreted factors that exhibited significant differential expression in only the $Gpdh1^{A10/B18}; Ldh^{16/17}$ double mutant. (C-F) Representative confocal images of upd3-GFP expression in the muscles of (C) $Gpdh1^{A10/+}; Ldh^{16/+}$ heterozygous controls, (D) $Gpdh1^{A10/B18}$ single mutants, (E) $Ldh^{16/17}$ single mutants, and (F) $Gpdh1^{A10/B18}; Ldh^{16/17}$ double mutants. The scale bar representing 50 µM in (C) applies to all other panels. (G) Quantification of upd3-GFP levels in the larval muscles. Data presented as a scatter plot with the lines representing the mean and standard deviation. P-values were calculated using an ANOVA followed by a Holm-Sidak test. *$P<0.05$.

signaling in the double mutant background would suppress the larval arrest phenotype. Indeed, triple mutants completed larval development and entered metamorphosis at a significantly higher rate than the double mutants (Fig 6A). Moreover, while we never observed a $Gpdh1^{A10/B18}; Ldh^{16/17}$ double mutant complete metamorphosis, a significant percentage of triple mutant pupae survived to adulthood (Fig 6B and 6C). We would note, however, that $upd3^{Δ}; Gpdh1^{A10/B18}; Ldh^{16/17}$ triple mutant larvae are of the same reduced size as double mutant controls (Fig 6D–6E), and that triple mutant adults were

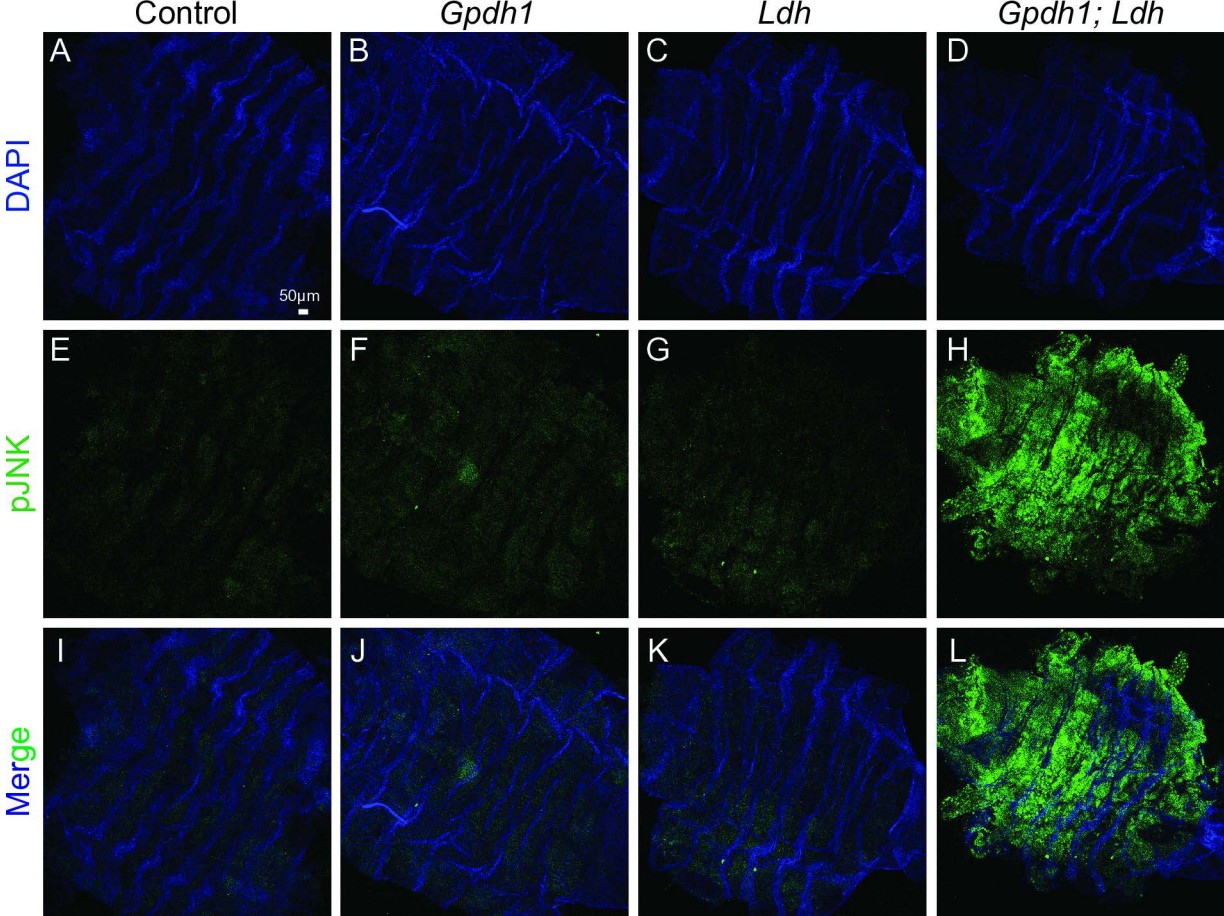

**Fig 4. *Gpdh1; Ldh* double mutants display elevated JNK activation in the larval muscles.** (A-L) Representative confocal images of anti-pJNK and DAPI in the larval muscles of (A,E,I) heterozygous controls (*Gpdh1^A10/+^; Ldh^16/+^*), (B,F,J) *Gpdh1^A10/B18^* single mutants, (C,G,K) *Ldh^16/17^* single mutants, and (D,H,L) *Gpdh1^A10/B18^; Ldh^16/17^* double mutants at 74-80 hrs after egg-laying. DAPI is shown in blue and anti-pJNK in green. The scale bar representing 50 µM in (C) applies to all other panels.

sick, and died within 2–3 days of eclosion, indicating that not all aspects of the *Gpdh1^A10/B18^; Ldh^16/17^* double mutant phenotype are regulated by Upd3.

We observed a similar, albeit weaker, phenomenon when we used *Mef2R-Gal4* to drive *upd3-RNAi* expression (abbreviated as *Mef2R-upd3-RNAi*) in the muscle of *Gpdh1^A10/B18^; Ldh^16/17^* double mutant larvae. Consistent with the triple mutant analysis, a significantly higher percentage of *Gpdh1^A10/B18^; Ldh^16/17^* double mutants expressing *Mef2R-upd3-RNAi* pupated by 8-days after egg-laying (AEL) relative to negative controls (Fig 7A). We would note, however, that *Mef2R-upd3-RNAi* did not ultimately increase the total number of double mutant larvae that pupated, as the percent of larvae that pupated 12-days after egg-laying was similar among all *Gpdh1^A10/B18^; Ldh^16/17^* double mutant strains (Fig 7B). Together, these findings suggest that *Mef2R-upd3-RNAi* accelerates the development of *Gpdh1^A10/B18^; Ldh^16/17^* double mutant larvae but does not enhance viability. Double mutants expressing *Mef2R-upd3-RNAi* also showed no increase in either body size (Fig 7C), or adult viability, as we never observed a double mutant adult in three independent experiments (6 vials per experiment; 20 larvae per vial). Together, these findings suggest that Upd3 expression from the muscle of *Gpdh1^A10/B18^; Ldh^16/17^* double mutants serves a role in slowing larval growth. However, considering that the *upd3* mutation is a stronger suppressor of

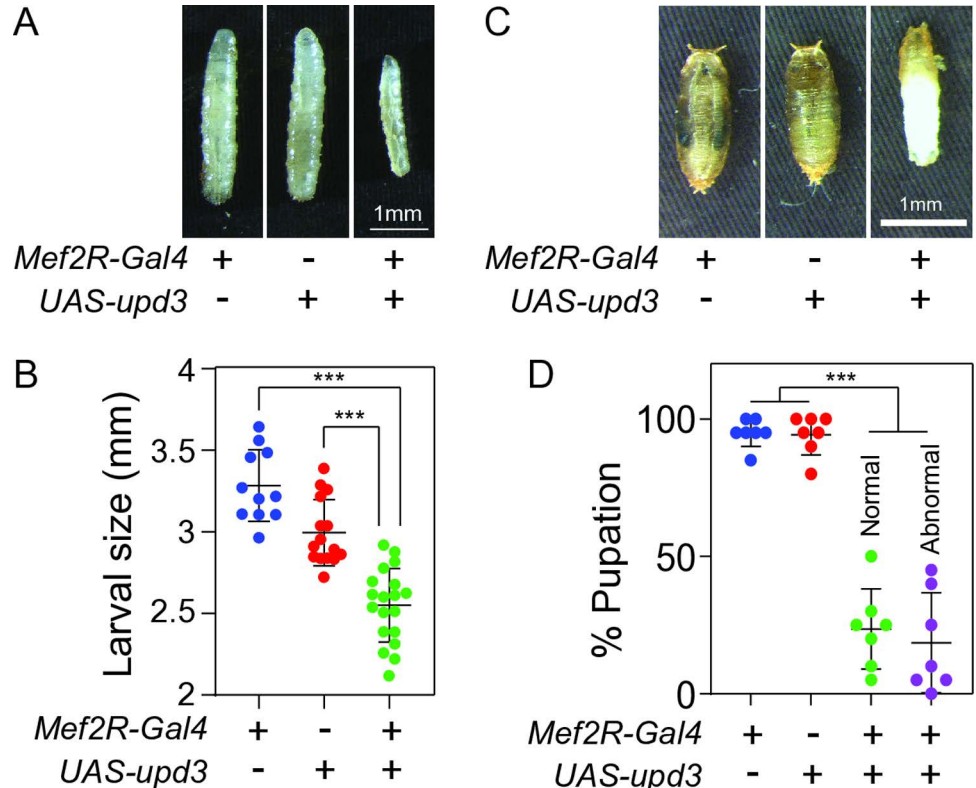

**Fig 5. Overexpression of Upd3 in larval muscle inhibits growth and disrupts pupal development.** (A) Representative larval images of the indicated genotypes 6 days after egg-laying (AEL). The scale bar represents 1 mm and applies to all panels. (B) Quantification of the larval size for the indicated genotypes at 6 days AEL. (C) Representative images of pupae from indicated genotypes at 12 days AEL. Note that *Mef2R-upd3* expression induces incomplete metamorphosis, as evident by the presence of larval cuticle over the posterior region. (D) The rate of pupation was quantified for larvae of the indicated genotypes 12 days AEL. For *Mef2R-upd3* expression, pupae with a normal morphological appearance were quantified separately from those with abnormal larval characteristics. For (B,D), data presented as a scatter plot with the lines representing the mean and standard deviation. *P*-values were calculated using an ANOVA followed by a Holm-Sidak test. \*\*\**P* < 0.001.

the *Gpdh1^{A10/B18}; Ldh^{16/17}* double mutant phenotypes than *Mef2R-upd3-RNAi*, we hypothesize that Upd3 expression from other tissues also contributes to the double mutant developmental defects.

### *upd3* mutations do not alter the metabolomic profile of *Gpdh1; Ldh* double mutants

One potential mechanism by which the *upd3* mutation could suppress the *Gpdh1^{A10/B18}; Ldh^{16/17}* larval arrest phenotype would be via activation of additional metabolic pathways that compensate for loss of Gpdh1 and Ldh activity. We examined this possibility by analyzing *Gpdh1^{A10/B18}; Ldh^{16/17}* double mutant larvae and *upd3^Δ; Gpdh1^{A10/B18}; Ldh^{16/17}* triple mutant larvae using a semi-targeted metabolomics approach (S3 Table). Our analysis revealed no significant metabolic differences between the two strains (S8 Fig). Notably, the levels of both lactate and glycerol-3-phosphate were unchanged in the triple mutant as compared with double mutant larvae (S8B and S8C Fig), indicating that the *upd3* mutation does not restore glycolytic metabolism in *Gpdh1^{A10/B18}; Ldh^{16/17}* double mutants. Overall, these results suggest that the developmental phenotypes displayed by *Gpdh1^{A10/B18}; Ldh^{16/17}* double mutants aren't simply the result of metabolic dysfunction but rather stem from an intentional larval arrest due, in part, to elevated Upd3 signaling.

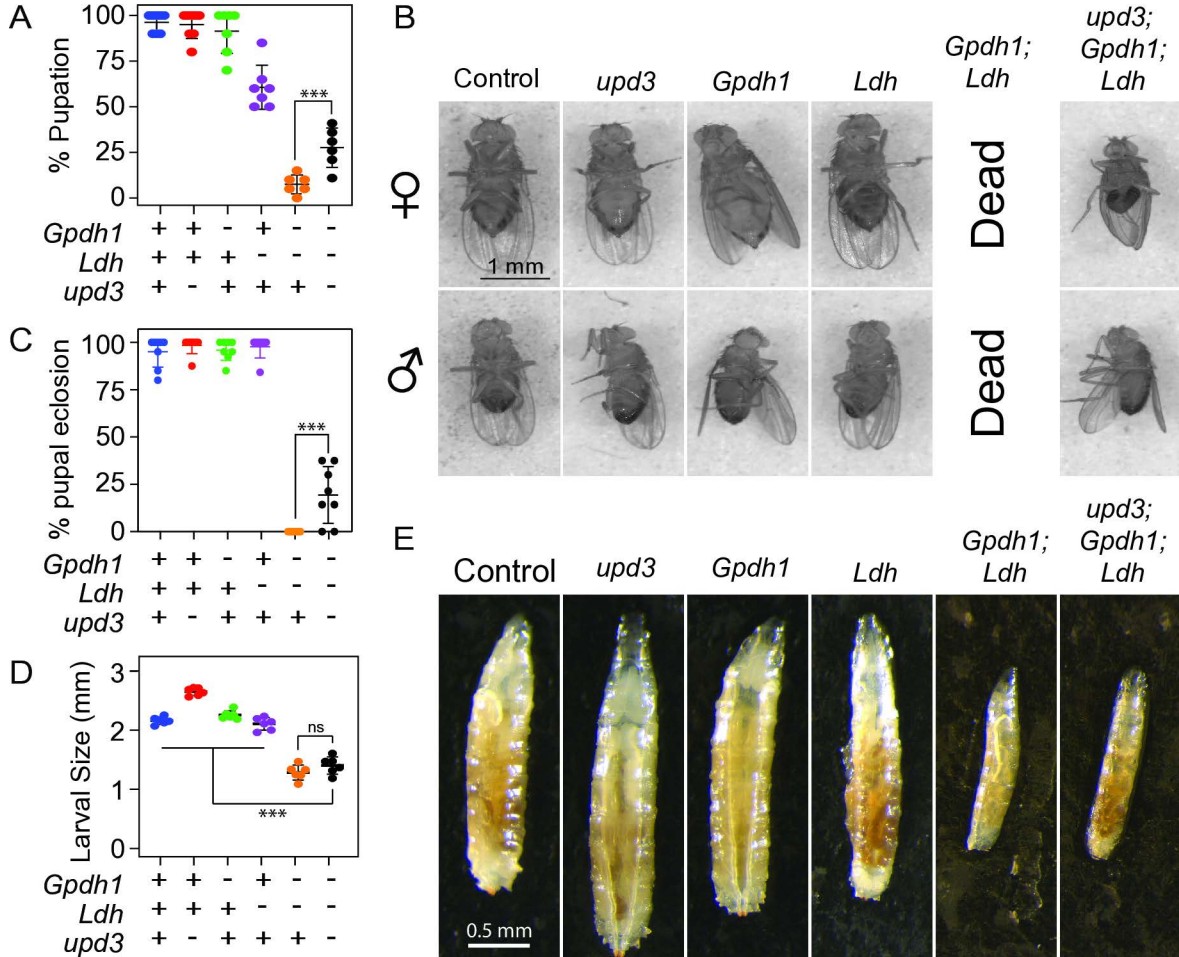

**Fig 6. *upd3* loss-of-function mutations suppress the synthetic lethal phenotype of *Gpdh1^{A10/B18}*; *Ldh^{16/17}* double mutants.** (A) The rate of pupation was quantified for larvae of the indicated genotypes 12 days after egg-laying (AEL). Triple mutant larvae (*upd3^Δ*; *Ldh^{16/17}*; *Gpdh1^{A10/B18}*) pupated at a significantly higher rate when compared with *Gpdh1^{A10/B18}*; *Ldh^{16/17}* double mutants. (B,C) A significant number of male and female triple mutant pupae successfully completed metamorphosis whereas all *Gpdh1^{A10/B18}*; *Ldh^{16/17}* pupae failed to eclose. (B) Representative images of adults from the control and triple mutant strains. No double mutant adults were observed in 3 independent experiments. (C) Quantification of pupal viability for the indicated genotypes. (D) Quantification of the larval size for the indicated genotypes 6 days AEL. There was no significant difference between the length of double and triple mutant larvae. (E) Representative images of the larvae measured in (D). For (B), the scale bars represent 1 mm and applies to all images in the respective panels. For (E), the scale bars represent 0.5 mm and applies to all images in the respective panels. Data presented in (A,C,D) as a scatter plot with the lines representing the mean and standard deviation. *P*-values were calculated using an ANOVA followed by a Holm-Sidak test. ****P* < 0.01.

## Loss of Gpdh1 and Ldh activates JAK/STAT signaling in the prothoracic gland

Considering that Upd3 is a secreted factor that activates JAK/STAT signaling in target cells [33], we used a *10XStat92E-GFP* reporter to identify tissues within *Gpdh1^{A10/B18}*; *Ldh^{16/17}* double mutant larvae that display increased JAK/STAT signaling [24,34]. Consistent with Upd3 serving a key role in limiting systemic growth, *Gpdh1^{A10/B18}*; *Ldh^{16/17}* double mutants exhibited widespread *10XStat92E-GFP* reporter expression when compared with either the heterozygous control strain, *Ldh^{16/17}* single mutant, or *Gpdh1^{A10/B18}* single mutant. Notably, we observed increased *10XStat92E-GFP* expression in the fat body (S9A–S9D Fig), salivary glands (S9E–S9H Fig), muscles (S9I–S9L Fig), and CNS (Fig 8A–8D). We also observed significantly increased *10XStat92E-GFP* expression in the PG of double mutants relative to the single mutants

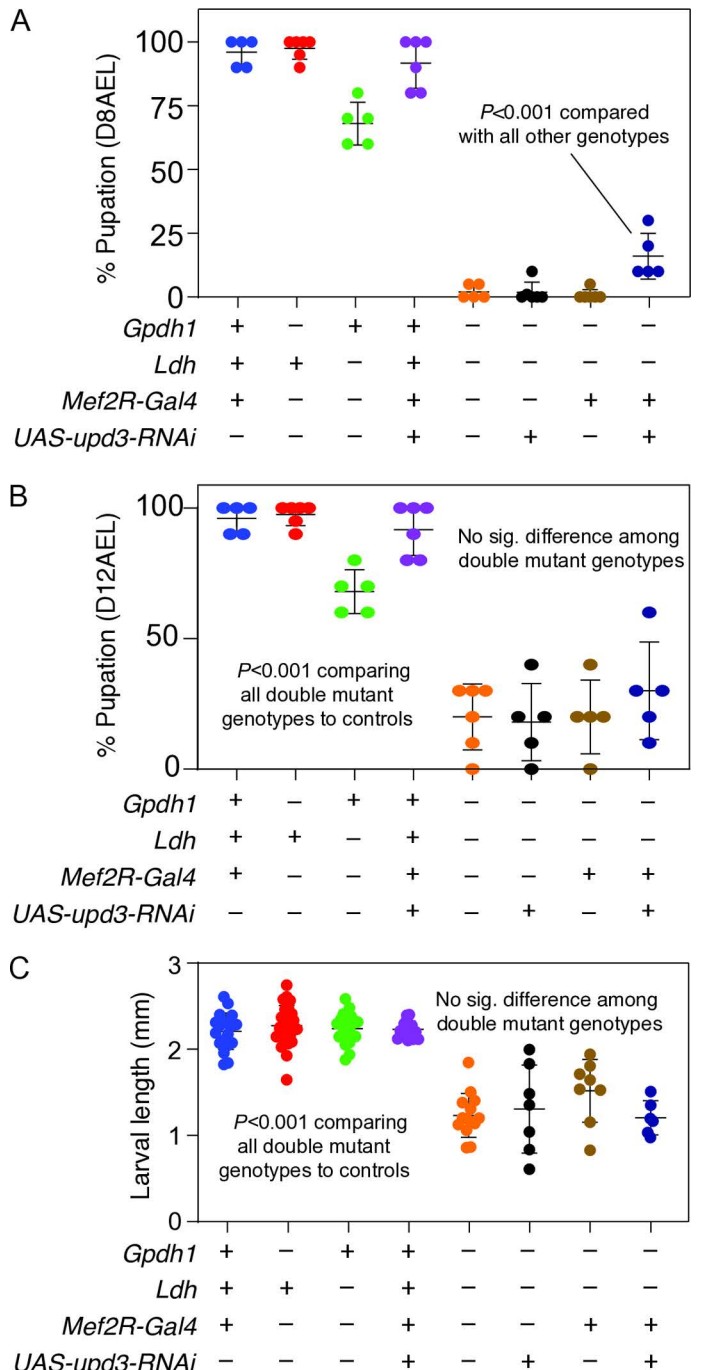

**Fig 7. RNAi targeting of *upd3* expression in muscle of *Gpdh1; Ldh* double mutant partially suppresses the developmental arrest phenotype.**
(A,B) The percentage of pupation was quantified for larvae of the indicated genotypes at (A) 8 days after egg-laying (AEL) and (B) 12 days AEL. (A) *Ldh16/17; Gpdh1A10/B18* double mutant larvae expressing *Mef2R-upd3-RNAi* exhibited significantly higher levels of pupation at 8 days AEL when compared with all double mutant controls. (B) By 12 days AEL, the number of pupae present within vials of *Ldh16/17; Gpdh1A10/B18* double mutant controls was similar to that found in vials of double mutants expressing *Mef2R-upd3-RNAi*. (C) Quantification of the larval size for the indicated genotypes. No significant difference in body length was observed between *Ldh16/17; Gpdh1A10/B18* double mutant larvae expressing *Mef2R-upd3-RNAi* and double mutant controls at 74-80 hrs AEL. Data presented as a scatter plot with the lines representing the mean and standard deviation. *P*-values were calculated using an ANOVA followed by a Holm-Sidak test. \*\*\**P* < 0.01.

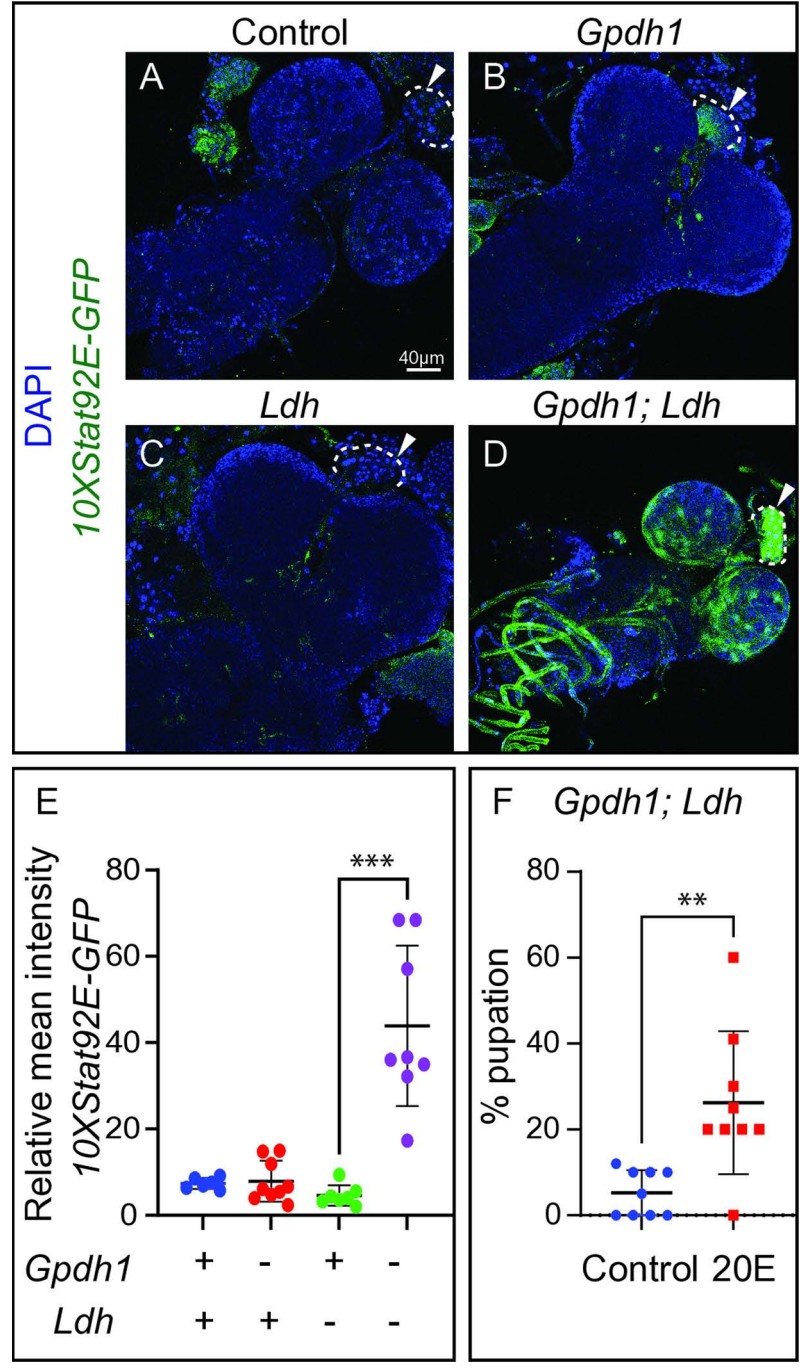

**Fig 8. 20E dietary supplementation partially suppresses the *Gpdh1; Ldh* double mutants larval arrest phenotype.** (A-D) Representative confocal images of *Stat-GFP* expression in the central nervous system and prothoracic gland (PG) of *Gpdh1^{A10/B18}; Ldh^{16/17}* double mutants as compared to the heterozygous control (*Gpdh1^{A10/+}; Ldh^{16/+}*) and single mutant strains (*Gpdh1^{A10/B18}* and *Ldh^{16/17}*) at 12 days after egg-laying (AEL). DAPI is shown in blue and *Stat-GFP* expression in green. The scale bar in (A) represents 40 μM and applies to (B-D). White dashed line marks the PG in panels (A-D). (E) Quantification of *Stat-GFP* expression in the PG of indicated genotypes at 74–80 hrs AEL. (F) A graph illustrating the percent of *Gpdh1^{A10/B18}; Ldh^{16/17}* double mutant that pupated when raised on yeast-molasses agar that contains either 20-hydroxyecdysone (20E) or the solvent (ethanol) control at 12 days AEL. (E,F) Data presented as a scatter plot with the lines representing the mean and standard deviation. (E) *P*-values were calculated using an ANOVA followed by a Holm-Sidak test. ***$P < 0.01$. (F) *P* value was calculated using a Mann-Whitney test. **$P < 0.01$.

(Fig 8A–8E), although, *Gpdh1*<sup>A10/B18</sup> single mutants displayed slightly elevated *10XStat92E-GFP* expression relative to the *Ldh*<sup>16/17</sup> single mutant and heterozygous control strain (Fig 8B). Together, these observations not only demonstrate that JAK/STAT signaling is widely activated upon simultaneous loss of Ldh and Gpdh1 but also raise the possibility that 20E signaling could be altered in *Gpdh1*<sup>A10/B18</sup>; *Ldh*<sup>16/17</sup> double mutants.

## Ecdysone feeding is sufficient to suppress the *Gpdh1; Ldh* larval lethal phenotype

A previous study of tumorous wing discs demonstrated that Upd3 signaling can suppress ecdysone production in the PG and inhibit larval development [21]. Thus, our observation that *10XStat92E-GFP* expression is elevated within the *Gpdh1*<sup>A10/B18</sup>; *Ldh*<sup>16/17</sup> PG raises the possibility that the double mutant larval arrest phenotype stems, in part, from decreased ecdysone signaling. Due to technical challenge of synchronizing *Gpdh1*<sup>A10/B18</sup>; *Ldh*<sup>16/17</sup> double mutant larvae, we were unable to accurately measure accurate 20E levels in this genetic background. However, we were able to use an LC-MS-based approach to demonstrate that 20E levels were significantly elevated in *upd3*<sup>Δ</sup> larvae (S10A Fig), supporting the proposed model wherein elevated Upd3 signaling results in decreased ecdysone levels [21]. We therefore further tested our hypothesis by supplementing the food of *Gpdh1*<sup>A10/B18</sup>; *Ldh*<sup>16/17</sup> double mutant larvae with 20E. Consistent with the proposed model, we found that dietary 20E supplementation significantly increased the pupation rate of *Gpdh1*<sup>A10/B18</sup>; *Ldh*<sup>16/17</sup> double mutants (Fig 8F) without affecting the pupation rate of single mutants (S10B Fig). However, 20E feeding did not restore pupal viability, as we never observed a adult eclose from double mutant cultures following 20E feeding (n = 6 vials of 20 larvae per vial; 3 independent experiments). One possible explanation for why 20E supplemented food does not rescue adult viability is that Upd3 also influences 20E signaling following the cessation of larval feed and onset of metamorphosis. Regardless, these results demonstrate that the *Gpdh1*<sup>A10/B18</sup>; *Ldh*<sup>16/17</sup> double mutant phenotype is sensitive to changes in 20E abundance. Moreover, these findings support a model in which simultaneous loss of Gpdh1 and Ldh results in elevated Upd3 expression, which in turn leads to JAK/STAT activation in the PG and decreased 20E signaling, ultimately leading to a larval arrest (Fig 9).

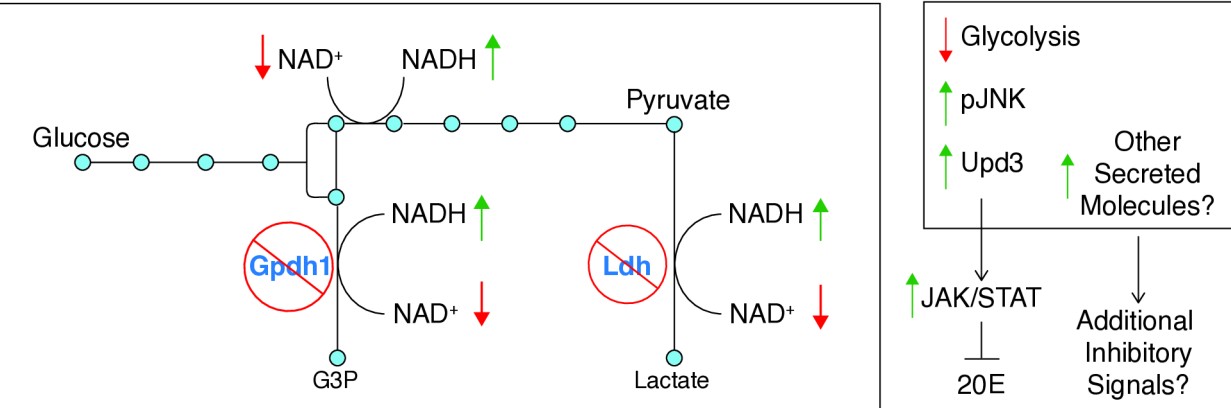

**Fig 9. A proposed model for how loss of Ldh and Gpdh1 activity nonautonomously restricts larval growth.** Loss of Gpdh1 and Ldh activity results in reductive stress, decreased glycolytic flux, and elevated *upd3* expression within the larval tissue. Increased Upd3 levels trigger systemic activation of JAK/STAT signaling in larval tissues, including in the PG. Consequently, 20E titers are reduced, thus impeding larval development. We would hypothesize, however, that additional factors are likely involved in the double mutant developmental phenotypes, as supported by the RNA-seq data and the fact that loss of Upd3 does not completely suppress the double mutant phenotypes. Moreover, some aspects of the developmental phenotypes are inevitably the result of metabolic limitations. Future studies will be required to disentangle the distinct phenotypic contributions of signal transduction cascades and metabolic bottlenecks within double mutant larvae.

## Discussion

Here we demonstrate that the developmental arrest displayed by the *Gpdh1; Ldh* double mutants extends beyond simple metabolic disruption and instead stems, in part, from changes in systemic growth factor signaling. Specifically, our findings reveal that loss of both Gpdh1 and Ldh activity in muscle correlates with increased JNK activation and elevated expression of the cytokine Upd3, a key regulator of the Jak/Stat signaling pathway that is capable of inhibiting larval growth. Our studies also demonstrate that this increase in Upd3 expression contributes to the arrested growth of these metabolically compromised larvae, as *upd3* loss-of-function mutations can partially suppress the developmental phenotypes of *Gpdh1; Ldh* double mutants. Moreover, our finding that 20E feeding can also suppress the larval arrest phenotype is consistent with previous observations that Upd3 inhibits ecdysone synthesis via JAK/STAT activation in the PG [21]. Overall, our findings suggest a mechanism by which disrupted glycolytic metabolism activates JNK signaling and Upd3 expression in muscle, which in turn inhibits larval development and 20E signaling (Fig 9).

While we have yet to identify the molecular mechanism that directly links loss of Gpdh1 and Ldh activity with increased JNK signaling and Upd3 expression, we hypothesize that this connection stems from the essential role that Ldh and Gpdh1 serve in maintaining the cytosolic NAD$^+$/NADH ratio (Fig 9). Loss of either Ldh or Gpdh1 activity results in a significant decrease in the NAD$^+$/NADH ratio [19]. Thus, double mutant larvae likely experience significant reductive stress, which has far-reaching consequences on cellular metabolism and physiology, including reduced glycolytic flux, disrupted redox signaling, and altered disulfide bond formation [35,36]. Moreover, while larvae in the wild are unlikely to experience conditions that specifically inhibit GPDH1 and LDH, our genetic system mimics the cytosolic redox challenges of hypoxia-exposed animals [35,36]. Considering that *Drosophila* larvae are exposed to low oxygen environments within their food [37], our studies are directly relevant towards understanding how systemic larval growth and development respond to reductive stress when challenged with hypoxic conditions.

In addition to the model presented above, our studies suggest that additional cell signaling mechanisms are active in *Gpdh1; Ldh* double mutants. As noted in the results, the *upd3* mutation only partially suppresses the double mutant phenotype, and while metabolic dysfunction, itself, likely contributes to the overall developmental defects, we suspect that other mechanisms are involved. In this regard, we would highlight a recent study of glycolysis in the larval fat body [38], which demonstrated that RNAi knockdown of the glycolytic enzyme Pglym78 within fat body cells induces systemic developmental defects, including growth delays and muscle wasting. Intriguingly, the phenotypes induced by depletion of Pglym78 within the fat body did not appear to result from elevated Upd3 signaling [38], but rather involves REPTOR-dependent changes in TNF-→/egr as well as the secreted protein ImpL2, which inhibits insulin signaling. When considered in light of our findings, this earlier study by Rodríguez-Vázquez indicates that a multitude of signaling pathways are likely coordinating cell- and tissue-specific changes in metabolic flux with systemic growth and development.

Beyond the relevance of our study in understanding the connection between glycolytic flux, reductive stress, and developmental progression, the ability of *Drosophila* larval development to tolerate the loss of two key enzymes involved in cytosolic NAD$^+$/NADH balance is quite intriguing. This finding highlights the exceptional metabolic robustness of *Drosophila* larval development and raises questions as to how larval growth proceeds in the face of such severe metabolic insults. We would note, however, that our observations are not without precedent. Classic studies of *COX5A* (also known as *tenured*) revealed that the loss of this electron transport chain subunit induces a specific eye development phenotype [39]. The defects induced by *COX5A* mutations are not simply the result of defects in ATP synthesis, but rather stem from the activation of a pathway involving AMPK and p53, which eliminates Cyclin E and induces a cell cycle arrest [39]. These foundational studies using *tenured* mutations were subsequently validated by two genetic screens, which demonstrated that depletion of electron transport chain subunits induce specific defects in eye development [40,41]. Our studies of Ldh and Gpdh1 are built upon these earlier observations and again demonstrate that *Drosophila* larvae experiencing severe metabolic disruptions don't simply die due to metabolic stress but rather arrest development through the activity of specific signaling pathways.

Conceptually, our findings offer an interesting mechanistic rationale for how animals in the wild delay development to survive temporary metabolic disruptions. Successful responses to environmental stressors such as starvation and hypoxia require both cell autonomous and nonautonomous responses that must be coordinated across all tissues. Similarly, animals also encounter environmental insults that affect individual tissues, including bacterial and parasitic infections, toxins that target specific organs, and nutrient imbalances that impinge upon cell-specific metabolic bottlenecks. Animal development has evolved to sense these tissue-specific metabolic stresses and produce signals from effected cells that are subsequently amplified and recognized by unaffected tissues, thus allowing development and metabolism to be altered in a coordinate manner. For example, *Drosophila* hemocytes respond to parasitic wasp infections by inducing a cell intrinsic increase in glycolytic metabolism that not only facilitates an effective immune response but also produces an adenosine signal that remodels peripheral metabolism [42–44].

Our findings that Upd3 and 20E function as part of a systemic response to tissue-specific metabolic disruptions provide new insights into how metabolic flux and developmental growth are precisely coordinated. Upd3 is a well-studied stress-induced cytokine that communicates cell-specific information with peripheral tissues [21,25,29,33,45–47], and our finding reinforces a previously proposed model in which elevated Upd3 signaling from the muscle induces a feedforward loop that inhibits 20E signaling and delays *Drosophila* development [21]. In this regard, the involvement of a steroid hormone is an important feature of the regulatory mechanism, as it allows for a single tissue to indirectly influence the developmental progression of entire animal.

These studies also highlight muscle as a key tissue for the systemic regulation of growth, development, and physiology. Our findings expand upon other recent studies in the fly which demonstrate that genetic alterations specifically in the muscle and motoneurons can induce systemic effects [48–50]. Similarly, glycolytic flux within mouse skeletal muscles is capable of regulating the metabolism of adipose tissue and dysfunctional metabolism within human skeletal muscle plays key role in the progression of Type 2 diabetes [51,52]. Central to this regulation are muscle-derived cytokines and metabolites (myometabolites) that are synthesized and released in response to stress [53,54]. Our findings highlight production of Upd3 from muscle as an attractive genetic model for future studies of muscle-derived signals.

Finally, the role of Upd3 in communicating tissue-specific metabolic stress with systemic growth provides yet another example of how members of the Unpaired (Upd) gene family serve to coordinate systemic responses with tissue-specific metabolic status. In addition to previous studies of Upd3 noted above, both Upd1 and Upd2 also serve key roles in interorgan communication. For example, Upd1 has been reported to be released from brain and to regulate obesity regulated behaviors [55]. Meanwhile, metabolic cues within fat body control secretion of the cytokine Upd2, which in turn regulates insulin secretion and synapse reorganization [56–58]. While neither Upd1 nor Upd2 expression was altered in our studies, we would note that the whole-body RNA-seq analysis used herein could very well overlook changes in the expression of these cytokines. Therefore, future studies should determine if Upd1 and Upd2 signaling is also disrupted in either the *Gpdh1; Ldh* double mutant or in other mutants with disrupted glycolytic flux. Moreover, while our study focused on the muscle, we hypothesize that elevated Upd3 in tissues besides the muscle also contribute to the *Gpdh1; Ldh* growth arrest – a hypothesis supported by our observation that *Mef2R-upd3-RNAi* is a less potent suppressor of the double mutant phenotypes than the whole body *upd3* knockout (i.e., the *upd3; Gpdh1; Ldh* triple mutant). We intend to examine this possibility in future studies.

Moving forward, the key question emerges of how metabolic stress regulates JNK and Upd3 signaling? Both individual metabolites (i.e., lactate and G3P) as well end products of cell-specific metabolic dysfunction (e.g., redox imbalance and ROS accumulation) represent potential signals that could trigger Upd3 signaling as well as related modes of interorgan communication. For example, lactate influences electron transport activity [59], Hif1→ stability [60], and histone modifications [61], all via mechanisms that are independent of its role in central carbon metabolism. Similarly, G3P not only acts in a mitochondrial electron shuttle [62], but the glycolytic precursor of G3P, dihydroxyacetone phosphate (DHAP), has recently emerged as an activator of mTor signaling [63]. Therefore, altered levels of either metabolite could indirectly

influence JNK/Upd3 signaling via multiple mechanisms. In this regard, it would be interesting to determine whether circulating lactate or G3P is capable of influencing JAK/STAT signaling in the PG – potentially providing a direct link between the metabolic status of peripheral tissues and 20E signaling. In addition, the reductive stress imposed upon cells by simultaneous loss of Ldh and Gpdh1 likely has substantial direct effects on signal transduction pathways, where changes in NADH concentration and diminished glycolytic flux, as well as the inevitable effects on disulfide bond formation can induce global changes in cell signaling [35,36]. Regardless of the answer, our study highlights that core metabolic enzymes are not simple products of "housekeeping genes," but rather serve dynamic cellular functions that are fully integrated into the signaling pathways that regulate multicellular growth and development.

## Methods

### *Drosophila melanogaster* husbandry and genetic analysis

Fly stocks were maintained at 25ºC on Bloomington Drosophila Stock Center (BDSC) food. Larvae were raised and collected as previously described [64]. Unless noted, *Ldh* mutant larvae harbored a trans-heterozygous combination of *Ldh$^{16}$* (RRID:BDSC_94698) and *Ldh$^{17}$* (RRID:BDSC_94699) [22], and *Gpdh1* mutant larvae harbored a trans-heterozygous combination of *Gpdh1$^{A10}$* (RRID:BDSC_94702) and *Gpdh1$^{B18}$* (RRID:BDSC_94703) [19]. The *upd3* mutant strain was obtained from the BDSC (RRID:BDSC_55728) [65]. The *Ldh-GFP$^{Genomic}$* stock (RRID:BDSC_94704) used for colocalization analysis of Ldh and Gpdh1 was previously described [20,66]. The *w$^{1118}$; P{w[+mC]=10XStat92E-GFP}2* stock was obtained from the BDSC (RRID:BDSC_26198) [34] and recombined with *Ldh$^{16}$* to visualize STAT expression levels in the tissues of (*Gpdh1$^{A10/B18}$; Ldh$^{16/17}$*) double mutant larvae. Muscle specific knockdown of Ldh was conducted using transgenes that express GAL4 in the muscle (P{w[+mC]=GAL4-Mef2.R}3; RRID:BDSC_27390). RNAi experiments were conducted using transgenes that targeted *Ldh* expression (RRID:BDSC_33640). The Gal4 line was placed into a *Gpdh1$^{B18}$* mutant background and crossed to *w$^{1118}$; Gpdh1$^{A10}$; UAS-Ldh-RNAi* to attain the muscle specific knockdown. The efficiency of *UAS-Ldh-RNAi* was demonstrated previously [22]. To confirm the expression of Gal4 in muscles, *Mef2R-Gal4* adults were crossed to Sco/CyO; P{y[+t7.7] w[+mC]=20XUAS- 6XmCherry-HA}attP2 (RRID:BDSC_97863). Knockdown of *Gpdh1* in the muscle was done by crossing the transgene that targeted *Gpdh1* expression (RRID:BDSC_51474) with *Mef2R-Gal4*. For muscle specific knockdown of *upd3* in double mutant larvae, *Mef2R-Gal4* was recombined with *Ldh$^{17}$* allele and *UAS-upd3-RNAi* (RRID:BDSC 32859) [67] was recombined with *Ldh$^{17}$* allele, to finally cross the following two stocks- *w$^{1118}$; Gpdh1$^{A10}$; Mef2R-Gal4, Ldh$^{17}$* with *w$^{1118}$; Gpdh1$^{B18}$; UAS-upd3-RNAi, Ldh$^{16}$*. The *UAS-upd3* transgenic strain used to overexpress Upd3 was a kind gift from Bruce Edgar's lab. The *w[*]; P{w[+mC]=upd3-GAL4.Z}2, P{UAS-GFP.U}2/CyO* stock was obtained from the BDSC (RRID:BDSC_ 98420) [25] and recombined with *Gpdh1$^{A10}$* to visualize Upd3 expression levels in the tissues of (*Gpdh1$^{A10/B18}$; Ldh$^{16/17}$*) double mutant larvae. Flybase was used as an essential reference tool throughout this study [68,69].

### Immunofluorescence

Larval tissues were dissected in 1X phosphate buffer saline (PBS; pH 7.0) and fixed with 4% paraformaldehyde in PBS for 30 minutes at room temperature. Fixed samples were subsequently washed once with PBS and twice with 0.3% PBT (1x PBS with Triton X-100) for 10 mins per wash.

For GFP antibody staining, fixed tissues were incubated with goat serum blocking buffer (4% Goat Serum, 0.3% PBT) for one hour at RT and stained overnight at 4 ºC with primary antibody Rabbit anti-GFP diluted 1:500 (#A11122 Thermo Fisher). Samples were washed three times using blocking buffer and stained with secondary antibody Alexa Fluor 488 Goat anti-Rabbit diluted 1:1000 (#R37116; Thermo Fisher) for either 4 hrs at room temperature or overnight at 4 ºC. Stained tissues were washed with 0.3% PBT, immersed in DAPI (0.5µg/ml 1X PBS) for 15 mins and then mounted with a vector shield with DAPI (Vector Laboratories; H-1200-10).

For pJNK staining, tissues were incubated in primary antibody anti-ACTIVE JNK pAb, Rabbit (Promega v7931) diluted 1:500 in blocking buffer (3% BSA in 1% PBT), and left for overnight at 4 ºC. The rest of the protocol was the same as used for anti-GFP staining.

Ldh protein expression was visualized using an anti-Ldh antibody (Boster bio DZ41222) as previously described [20]. For larval tissues staining with anti-Gpdh1 antibody (Boster Bio DZ41223), the same protocol was used as for anti-Ldh staining, except that the anti-Gpdh1 antibody was diluted 1:100.

### Mounting and imaging

All the stained tissues were mounted using VECTASHIELD containing DAPI (Vector Laboratories; H-1200–10). Multiple Z-scans of individual tissues were acquired using the Leica SP8 confocal microscope in the Light Microscopy Imaging Center at Indiana University, Bloomington. Unless noted, figure panels contain a representative Z-scan. For presenting the Z stack in a 2D image, the Z projection tool of Fiji was used, with maximum projection. Image acquisition and analysis settings across groups in each experiment were the same.

Quantification of Gpdh1 and Ldh antibody staining was conducted by measuring the mean intensities for Gpdh1, Ldh and DAPI in the region of interest (ROI) per image, using Fiji. Gpdh1 and Ldh mean intensities were normalized to the mean DAPI intensity for the same ROI per image and plotted by using GraphPad Prism v10.1. Statistical analysis was conducted as described in figure legends.

### Imaging and quantification of larval size

Third instar larvae were collected and imaged using Leica MZ 10F microscope. The size of the larvae was measured by drawing a line from posterior to anterior end and noting down the total length by using Fiji. Data was plotted by using GraphPad Prism v10.1. Statistical analysis was as described in figure legends.

### Adult imaging

Third instar wandering larvae were kept on a molasses agar plate with yeast paste. Starting on D9 after egg laying, pupae were checked for eclosion, and the newly eclosed adults were collected into a BDSC food vial. Adults were imaged by using Leica DFC 450.

### Percentage pupation

The pupation rate was calculated by placing 20 synchronized L1 larvae of each genotype on a molasses agar plate with yeast paste. Starting on day 4 after egg laying, plates were monitored every 24 hours for pupae until 12 days after egg-laying.

### Sample collection and extraction for Metabolomics Analysis

Mutant and control larvae were collected at the mid-L2 stage (~60 hours after egg-laying at 25ºC) and extracted as previously described [70]. Briefly, collected larvae were placed into a prechilled 1.5 mL centrifuge tube on ice, washed with ice-cold water, and immediately frozen in liquid nitrogen. For extraction, larvae were transferred to pretared 1.4 mm bead tubes, the mass was measured using a Mettler Toledo XS105 balance, and the sample was returned to a liquid nitrogen bath prior to being stored at -80°C. Samples were subsequently homogenized in 0.8 mL of prechilled (-20 °C) 90% methanol containing 2 μg/mL succinic-d4 acid, for 30 seconds at 6.45 m/s using a bead mill homogenizer located in a 4°C temperature control room. The homogenized samples were incubated at -20 °C for 1 hour and then centrifuged at 20,000 × g for 5 min at 4°C. The resulting supernatant was sent for metabolomics analysis at the University of Colorado Anschutz Medical Campus, as previously described [71].

## Statistical analysis of metabolite data

The metabolomics dataset was analyzed using Metaboanalyst version 6.0 [72] with data normalized to sample mass and preprocessed using log normalization and Pareto scaling. Graphs for the relative levels of individual metabolites were plotted by using GraphPad Prism v10.1. Statistical analysis for these graphs were conducted using a Mann-Whitney test.

## RNAseq Analysis

Mutant and genetically-matched control larvae were collected at the mid-L2 stage (~60 hours after egg-laying at 25ºC). Three biological replicates containing 25 L2 larvae were collected for each genotype and washed twice with ice-cold 1X PBS. Total RNA was extracted from homogenized samples using a RNeasy kit (Qiagen; 74104). Sequencing and data analysis was conducted in the Indiana University Center for Genomics and Bioinformatics.

Quality control of reads performed with FastQC v0.11.9 (https://www.bioinformatics.babraham.ac.uk/projects/fastqc/). Read quantification to transcripts was accomplished with Salmon v1.4.0 [73] using the ENSEMBL D. melanogaster BDGP6.28 cDNA sequences file, and the softmasked toplevel ENSEMBL BDGP6.28 genome assembly as the decoy sequences. Reads were imported into R and aggregated into gene counts using tximport v1.18.0 [74], and differential expression was performed using DESeq2 v1.30.0 [75].

## 20-hydroxyecdysone measurement

Mutant and control larvae (n = 30) were collected at the mid-L2 stage (~60 hours after egg-laying at 25ºC). Collected larvae were placed into a prechilled 1.5 mL centrifuge tube on ice, washed with ice-cold water, and immediately frozen in liquid nitrogen. Frozen larvae were transferred to pretared 1.4 mm bead tubes, the mass was measured using a Mettler Toledo XS105 balance, and the sample was returned to a liquid nitrogen bath prior to being stored at -80°C. Samples were subsequently homogenized in 0.8 mL of prechilled (-20 °C) 80% ethanol for 30 seconds at 6.45 m/s using a bead mill homogenizer located in a 4°C temperature control room. The homogenized samples were centrifuged at 20,000 × g for 5 min at 4°C. The resulting supernatant (600 μL) was transferred to a fresh eppendorf tube and purified by adsorption on a Sep-Pak C18 1 cc Vac Cartridge (Waters, WAT- WAT054955), washed with 2 mL of 0.1% aqueous formic acid and eluted with 2 mL of 70% aqueous acetonitrile. Elutes were vacuum dried and dried extracts were reconstituted in 100 μL of 200 proof ethanol, vortexed for 10 seconds and then centrifuged for 15 minutes at 15,000 x g at 4°C. 80 μL of supernatant was taken for analysis and placed in autosampler vials with low-volume inserts. Samples were analyzed for 20-hydroxyecdysone content by LC-MS/MS utilizing a 1290 Infinity II LC system from Agilent Technologies consisting of a Multisampler, thermostatted column compartment, and UHPLC binary pump coupled to an AB Sciex 4000 QTrap operated in MRM mode.

Samples were separated on an Agilent 2.1 x 50 mm EclipsePlus C18 RRHD UHPLC column with 0.1% aqueous acetic acid as mobile phase A and 0.1% acetic acid in acetonitrile as mobile phase B. Flow rate was 0.3 mL/min and the column compartment was held at 40°C. Solvent composition was held at 10% B for 1 minute, then linearly ramped to 90% B over 4 minutes, held at 90% B for 1 minute and then rapidly returned to starting conditions and held for 6 minutes. The sample injection consisted of 5 μL of sample plus 15 μL of water for improved peak shape.

The QTrap 4000 was operated with the Turbo V ion source in ESI positive ion MRM mode. 20HE was detected as the protonated molecule and then fragmented in the collision cell leading to product ions at m/z 165.2 and m/z 371.3. These two transitions were both monitored with 50 ms dwell times to ensure accurate peak shape.

## 20-hydroxyecdysone feeding

Larvae were fed 20-hydroxyecdysone (20E) in a modified molasses agar food. The exposure media was made by adding 115 mL of molasses and 24 grams of agar to 750 mL of boiling water on a stirring hot plate. The temperature was then

reduced to 60ºC, and 5 grams of yeast was added to the media, at which point 2 mL of media was pipetted into a 35 mm petri dish. The low agar concentration resulted in a semi-solid media that was suitable for diffusion of a 20E stock solution.

A 30 mM 20E (Sigma H5142) stock solution was prepared in 90% ethanol and 3.33 µL of stock solution was added to the modified molasses food, resulting in a final concentration of 50 µM [76,77]. Larvae were transferred to the 20E-supplemented food plates immediately after hatching and subsequently transferred to fresh 20E-supplemented plates every 24 hrs.

## Supporting information

**S1 Fig. Ldh and Gpdh1 expression patterns in the CNS, fat body, salivary gland, and intestine.** Representative confocal images of L2 larval tissues expressing *Ldh-GFP^Genomic* and immuno-stained with αGPDH1 antibody. DAPI is shown in blue, Ldh-GFP and Gpdh1 are represented in green and magenta, respectively. The rightmost panel displays the merged images of Ldh-GFP and Gpdh1 staining. (A-D) Intestine; (E-H) Dorsal side of CNS; (I-L) fat body; (M-P) salivary gland. Scale bars in leftmost panels represents 40 µM and applies to all other panels in the same row.
(PDF)

**S2 Fig. Characterization of either Ldh or Gpdh1 expression in salivary glands following loss of the other enzyme.** Quantification of Ldh and Gpdh1 expression in the salivary glands of the heterozygous control strain (*Gpdh1^A10/+^; Ldh^16/+*) and each single mutant strain (*Gpdh1^A10/B18*^ and *Ldh^16/17*^). (A-F) Representative confocal images of (A-C) Ldh expression and (D-F) DAPI staining in all three genotypes. (G) Ldh staining was quantified in the salivary glands of all three genotypes. (H-M) Representative confocal images of (H-J) Gpdh1 expression and (K-M) DAPI staining in all three genotypes. (N) Gpdh1 staining was quantified in the salivary glands of all three genotypes. The scale bar in all images represents 40 µM. The scale bar in (A) applies to (B-F) and the scale bar in (H) applies to (I-M). (G, N) All experiments are repeated a minimum of three times. n > 4 biological replicates. Data presented as a scatter plot with the lines representing the mean and standard deviation. *P*-values were calculated using an ANOVA followed by a Holm-Sidak test.**$P < 0.01$. *$P < 0.05$.
(PDF)

**S3 Fig. Characterization of either Ldh or Gpdh1 expression in larval fat body following loss of the other enzyme.** Quantification of Ldh and Gpdh1 expression in the larval fat body of the heterozygous control strain (*Gpdh1^A10/+^; Ldh^16/+*) and each single mutant strain (*Gpdh1^A10/B18*^ and *Ldh^16/17*^). (A-F) Representative confocal images of (A-C) Ldh expression and (D-F) DAPI staining in all three genotypes. (G) Ldh staining was quantified in the fat body of all three genotypes. (H-M) Representative confocal images of (H-J) Gpdh1 expression and (K-M) DAPI staining in all three genotypes. (N) Gpdh1 staining was quantified in the larval fat body of all three genotypes. The scale bar in all images represents 40 µM. The scale bar in (A) applies to (B-F) and the scale bar in (H) applies to (I-M). (G, N) Data presented as a scatter plot with the lines representing the mean and standard deviation. *P*-values were calculated using an ANOVA followed by a Holm-Sidak test. ***$P < 0.01$.
(PDF)

**S4 Fig. *Mef2R-Gal4* driven expression of 20XUAS-6XmCherry in larval tissues.** Representative confocal images of mCherry expression in (A-B) larval muscles, (C-D) gut muscles, (E-F) CNS, (G-H) fat body, and (I-J) salivary gland at 74–80 hrs after egg-laying. DAPI is shown in blue and *Mef2R-Gal4* driven 20XUAS-6XmCherry expression in red. The scale bars represent 50 µM.
(PDF)

**S5 Fig. The transgene to knockdown *Gpdh1* does not significantly reduce Gpdh1 levels in the muscles.** Representative confocal images of muscles 74–80 hrs after egg-laying from (A) control, (B) *Gpdh1^-^*, (C) *Mef2R-Gal4* driven *UAS-Gpdh1-RNAi* stained with *anti-Gpdh1* antibody. The scale bars represent 50 µM. (D) Quantification of the mean

intensity of Gpdh1 in all conditions. Data presented as a scatter plot with the lines representing the mean and standard deviation. *P*-values were calculated using an ANOVA followed by a Holm-Sidak test. *$P<0.05$.
(PDF)

**S6 Fig.  RNA-seq analysis of *Ldh* mutants, *Gpdh1* mutants, and *Gpdh1, Ldh* double mutants.** Volcano plot depicting the transcriptomic profiles of (A) *Ldh* mutants (*Ldh*[16/17]), (B) *Gpdh1* mutants (*Gpdh1*[A10/B18]), and (C) double mutants (*Gpdh1*[A10/B18]; *Ldh*[16/17]) relative to the respective heterozygous control strains. n = 3 biological replicates analyzed per genotype. Each sample contained 20 mid-L2 larvae. Vertical axis indicates -log10(FDR) and horizontal axis represents log (FC). The significantly upregulated genes are shown in yellow and downregulated are shown in black. FDR- fold discovery rate and FC-fold change.
(PDF)

**S7 Fig.  Quantification of pJNK intensity in the muscles.** Samples represented in Fig 4 were quantified for pJNK staining intensity. Data presented as a scatter plot with the lines representing the mean and standard deviation. *P*-values were calculated using an ANOVA followed by a Holm-Sidak test.  ***$P<0.001$.
(PDF)

**S8 Fig.  *upd3; Gpdh1; Ldh* triple mutants exhibit a metabolomic profile that is indistinguishable from *Gpdh1; Ldh* double mutants.** (A) A volcano plot showing no significant difference between the metabolomes of *Gpdh1; Ldh* double mutants and *upd3; Gpdh1; Ldh* triple mutants. (B-C) Relative abundance of (B) lactate and (C) glycerol-3-phosphate in the indicated genotypes. n = 6 biological replicates were collected from independent populations with 25 mid L2 larvae per sample. The experiment was repeated twice. Error bars represent standard deviation. No significant differences (ns) were observed in in double mutants as compared with the triple mutants. *P*-value was calculated by using the Mann-Whitney test. (D) Heatmap showing the top 25 metabolites rank order by significance among four genotypes - heterozygous control (HC), *upd3*[Δ] (U), *Gpdh1*[A10/B18]; *Ldh*[16/17] double mutant (DM) and the *upd3Δ; Gpdh1*[A10/B18]; *Ldh*[16/17] triple mutants (TM).
(PDF)

**S9 Fig.  *Stat-GFP* expression is increased in larval tissues of *Gpdh1; Ldh* double mutants.** (A-L) Representative confocal images of (A-D) fat body, (E-H) salivary glands and (I-L) muscles showing *Stat-GFP* expression in control, *Gpdh1*[A10/B18], *Ldh*[16/17] and *Gpdh1*[A10/B18]; *Ldh*[16/17] double mutants at 74–80 hrs AEL. The scale bar represents 40 μM. The scale bar in (A) applies to (B-L). (M-O) Quantification of the relative mean intensity (RMI) of *Stat-GFP* in fat body (M), salivary glands (N) and muscles (O). Data presented as a scatter plot with the lines representing the mean and standard deviation. *P*-values were calculated using an ANOVA followed by a Holm-Sidak test. **$P<0.01$.
(PDF)

**S10 Fig.  Loss of Upd3 increases 20E and dietary supplementation with 20E has no effect on *Gpdh1* and *Ldh* single mutant larval growth.** (A) A graph showing the increased level of 20E in *upd3*[Δ] mutants at the mid-L2 stage (60–66 hrs after egg-laying). (B) A graph illustrating the percent of *Gpdh1*[A10/B18] and *Ldh*[16/17] single mutants that pupated when raised on yeast-molasses agar that contains either ecdysone or the solvent (ethanol) control. All experiments are repeated a minimum of three times. n = 9 biological replicates. Data presented as a scatter plot with the lines representing the mean and standard deviation. *P*-values were calculated using an ANOVA followed by a Holm-Sidak test. *$P<0.05$.
(PDF)

**S1 Table.  RNA-seq analysis of *Gpdh1*[A10/B18] single mutant larvae relative to *Gpdh1*[A10/+] heterozygous controls, *Ldh*[16/17] single mutant larvae relative to the *Ldh*[16/+] heterozygous controls, and *Gpdh1*[A10/B18]; *Ldh*[16/17] double mutant larvae relative to *Gpdh1*[A10/+]; *Ldh*[16/+] heterozygous controls.**
(XLSX)

**S2 Table. A list of the genes that are significantly downregulated or upregulated in *Gpdh1*$^{A10/B18}$*; Ldh*$^{16/17}$ double mutant larvae relative to *Gpdh1*$^{A10/+}$*; Ldh*$^{16/+}$ heterozygous controls but are expressed at normal levels in single mutant larvae.**
(XLSX)

**S3 Table. Metabolomic analysis of *Gpdh1*$^{A10/+}$*; Ldh*$^{16/+}$ heterozygous controls (HC)*, upd3*$^{\Delta}$ single mutants (U), *Gpdh1*$^{A10/B18}$*; Ldh*$^{16/17}$ double mutant (DM), and *upd*$^{\Delta}$*; Gpdh1*$^{A10/B18}$*; Ldh*$^{16/17}$ triple mutants (TM).**
(XLSX)

**S4 Table. Raw data and summary statistics for all graphs.** The data for every graph in both the main text and supplementary material is listed within individual sheets. Sheets are labeled by the Figure number and panel.
(XLSX)

## Acknowledgments

We thank the Bloomington *Drosophila* Stock Center (NIH P40OD018537) for providing fly stocks, the *Drosophila* Genomics Resource Center (NIH 2P40OD010949) for genomic reagents, Flybase (NIH 5U41HG000739), the Indiana University Light Microscopy Imaging Center, the Indiana University Center for Genomics and Bioinformatics, and the Indiana University Mass Spectrometry Facility. Thanks to Shefali Shefali for critical comments on the manuscript.

## Author contributions

**Conceptualization:** Madhulika Rai, Jason M. Tennessen.

**Data curation:** Madhulika Rai, Hongde Li, Robert A. Policastro, Robert Pepin, Gabriel E. Zentner, Travis Nemkov, Jason M. Tennessen.

**Formal analysis:** Madhulika Rai, Hongde Li, Robert A. Policastro, Robert Pepin, Gabriel E. Zentner, Travis Nemkov, Jason M. Tennessen.

**Funding acquisition:** Jason M. Tennessen.

**Investigation:** Madhulika Rai, Hongde Li, Jason M. Tennessen.

**Methodology:** Madhulika Rai, Robert Pepin, Travis Nemkov, Angelo D'Alessandro, Jason M. Tennessen.

**Project administration:** Jason M. Tennessen.

**Resources:** Robert Pepin, Jason M. Tennessen.

**Supervision:** Jason M. Tennessen.

**Validation:** Jason M. Tennessen.

**Visualization:** Madhulika Rai, Jason M. Tennessen.

**Writing – original draft:** Madhulika Rai, Jason M. Tennessen.

**Writing – review & editing:** Madhulika Rai, Hongde Li, Robert A. Policastro, Robert Pepin, Travis Nemkov, Angelo D'Alessandro, Jason M. Tennessen.

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
