## [Decision Letter · Decision Letter 0]

5 Jan 2025

PGENETICS-D-24-01432

Glycolytic disruption restricts Drosophila melanogaster larval growth via the cytokine Upd3

PLOS Genetics

Dear Dr. Tennessen,

Thank you for submitting your manuscript to PLOS Genetics. After careful consideration, we feel that it has merit but does not fully meet PLOS Genetics's publication criteria as it currently stands. Therefore, we invite you to submit a revised version of the manuscript that addresses the points raised during the review process. As you will note, two reviewers raise mainly relevant technical points while one reviewer (#3) hopes to see more conceptual advance in the revised form. Addressing these points will likely require additional experimentation.

Please submit your revised manuscript within 60 days Mar 06 2025 11:59PM. If you will need more time than this to complete your revisions, please reply to this message or contact the journal office at plosgenetics@plos.org. Please include the following items when submitting your revised manuscript:

We look forward to receiving your revised manuscript.

Kind regards,

Ville Hietakangas

Academic Editor

PLOS Genetics

Pablo Wappner

Section Editor

PLOS Genetics

Aimée Dudley

Editor-in-Chief

PLOS Genetics

Anne Goriely

Editor-in-Chief

PLOS Genetics

**Journal Requirements:**

1) Please provide an Author Summary. This should appear in your manuscript between the Abstract (if applicable) and the Introduction, and should be 150-200 words long. The aim should be to make your findings accessible to a wide audience that includes both scientists and non-scientists. Sample summaries can be found on our website under Submission Guidelines:

https://journals.plos.org/plosgenetics/s/submission-guidelines#loc-parts-of-a-submission

3) Some material included in your submission may be copyrighted. According to PLOSu2019s copyright policy, authors who use figures or other material (e.g., graphics, clipart, maps) from another author or copyright holder must demonstrate or obtain permission to publish this material under the Creative Commons Attribution 4.0 International (CC BY 4.0) License used by PLOS journals. Please closely review the details of PLOSu2019s copyright requirements here: PLOS Licenses and Copyright. If you need to request permissions from a copyright holder, you may use PLOS's Copyright Content Permission form.

Potential Copyright Issues:

i) Please confirm (a) that you are the photographer of 2A, 4A, and 4C, or (b) provide written permission from the photographer to publish the photo(s) under our CC BY 4.0 license.

4) Please amend your detailed Financial Disclosure statement. This is published with the article. It must therefore be completed in full sentences and contain the exact wording you wish to be published.

**Reviewers' comments:**

Reviewer's Responses to Questions

Reviewer #1: The study stems from the intriguing finding that disrupting glycolytic activity in the larval muscle, does not only affect growth locally, but induces a systemic growth defect and developmental arrest. Following this observation, the authors finds that the developmental arrest induced in the Gpdh1; Ldh double mutant animals leads to increased expression of a cytokine Upd3 and elevated Jak/Stat receptor activity in the prothoracic gland of the CNS. Furthermore, the authors show that loss of Upd3, as well as exogenous ecdysone administration, can rescue the developmental defects caused by Gpdh1; Ldh double mutant. Based on these findings, the authors suggest a model where disruption of glycolytic activity in the muscle, result into cellular stress and growth defect, which is then coordinated at the systemic level through Upd3 and ecdysone signaling.

While the mechanism, Upd3 signal to prothoracic gland, has been already shown elsewhere (https://doi.org/10.1016/j.cub.2021.01.080), the study of Rai et al. demonstrate its utilization in an unexpected context. Indeed, Rai et al. tackles important, and still poorly understood, questions of how central metabolism is interconnected with cellular signaling pathways, and how animal growth is physiologically coordinated in general. Although many underlying mechanical details are left unanswered, the work of Rai et al. significantly improves our understanding of these topical questions. Overall, the manuscript is very well written, the methods and the data are clearly presented, and the data is supporting the conclusions. Therefore, I find this study to be important, and interesting to the readers of PLOS Genetics. However, to make some of the key findings more convincing, I would ask the authors to add few more control experiments to the manuscript. Please find my specific comments and suggestion figure by figure below.

Figure 1.

Please change aGPDH to αGPDH.

Figure 2.

The data in figure 2 are central to the study since it shows the tissue specificity of glycolytic disturbance to systemic growth. The data presented is clear, however it relies on single Ldh RNAi line. Since there is no apparent phenotype for to judge the functionality of this RNAi line, the reader is left wondering its efficiency. Thus, to make the story more convincing I kindly ask the authors to add either (1) functional confirmation of the used RNAi line, (2) the results of using an independent Ldh RNAi line, or preferentially (if feasible) (3) use of a reciprocal strategy with Ldh mutant combined with Gpdh knockdown in the muscle.

Figure 3.

Please mark the highlighted genes in Figure 3 to the volcano plots in Figure S5.

The authors performed RNAseq experiment to reveal transcriptional changes in Gpdh1; Ldh double mutant larvae. They identify the cytokine Upd3 as a possible candidate for regulating the systemic growth effect. Since the RNAseq experiments are performed from whole animals, it is left unanswered if Upd3 expression is indeed affected in the muscle. This is important especially in the light of the modest change in Upd3 expression (LogFC 1.3). To make their conclusion solid, the authors should provide some evidence that Upd3 expression is affected specifically in the larval muscle in the Gpdh1; Ldh double mutant animals.

The morphology of the double mutant muscle (panel F) seems very different from the single mutant controls. Could the authors comment this?

Figure 4.

The rescue of the Gpdh1; Ldh double mutant by Upd3 mutation is remarkable, and the lack of detectable metabolic restoration supports the hypothesis that Upd3 works through a different mechanism. This is a very nice experiment indeed.

Please indicate in the figure panel:

A, B & D: At which time point are these measurements?

Figure 5.

Panels A-D: Please indicate how consistent this result is? The reader is especially interested in the PG Stat92E-GFP expression.

Please add a description of the “white dashed line indicating PG” in panels A-D (to the figure legend).

The finding that the Gpdh1; Ldh double mutant developmental arrest is partially rescued by 20E feeding is striking. The reader is, however, left puzzled how specific this is to the genotype, or if extra 20E has a more general beneficial effect to larval growth. Therefore, I would suggest adding control genotypes, such as the Gpdh1 and Ldh single mutants to the panel E. In addition, please indicate if the 20E feeding led to any viable adults.

Minor corrections/comments:

Line 130 “many cells the CNS” to “many cells of the CNS”

Line 132 “high levels Gpdh1” to “high levels of Gpdh1”

Line 134 Please add reference here.

Line 228 Please remove “significantly” since no quantification is done here.

Lines 255-257 The authors avoid mentioning the specific tissue, muscle, when describing their model. Is there a specific reason for that? To me, the tissue of the metabolic disturbance, and the source of Upd3, seems especially revealing. Could the authors comment on this issue, and perhaps mention it in the discussion?

Lines 322-327 The description of using tissue specific Ldh RNAi is confusing. Could the authors correct it?

Line 460 Please omit “The scale bar in (A) applies to all panels”

Line 574 Omit “in”

Reviewer #2: Summary

In this manuscript, Rai et al provide evidence that the larval arrest phenotype observed in Ldh;Gpdh1 double mutants is not likely due to effects on metabolism, but instead is regulated in part by growth factor signaling. While the manuscript is well written and data are solid, the authors should provide stronger rationale in some cases. Additionally, quantification of IF data can make some conclusions stronger. Overall, this work advances understanding of regulation of larval growth during Drosophila development.

Major concerns

Line 190-191: The focal plane for the double mutant seems to be different from that shown for the control and single mutants. Since the pJNK signal is nuclear, the control and single mutant representative images should be at a focal plane that contains more nuclei. On a related note, quantification and/or showing more than one image for each (in a supplemental figure) would strengthen this result.

While the authors propose that the larval arrest phenotype in Ldh;Gpdh1 double mutants is due to defects in systemic growth factor signaling, they did not examine signaling pathways known to control larval growth, namely insulin/insulin-like growth factor or mTOR-mediated signaling. To provide more direct support of their hypothesis, the activity of these pathways in all genetic backgrounds (i.e., single mutants, double mutants, single mutants with RNAi of the other in a specific tissue, triple mutants, etc) should be examined.

In all cases for which the authors make conclusions based on immunofluorescence microscopy, the data could be strengthened by quantification and/or showing more than one image for each condition to show the range observed.

Lines 357-359: The uniformity, or not, of image acquisition settings across groups in a given experiment needs to be mentioned.

Moderate concerns

Line 139-140: Perhaps the statement “…we find no evidence to support this hypothesis” should not be so definite since Ldh and Gpdh1 expression was not examine in all tissues; the figures do not show, and the text does not mention, expression in Malpighian tubules or muscle in each of the single mutants.

Line 151: Efficiency of Ldh RNAi-mediated knockdown is not mentioned. If this is shown in a previous manuscript, please provide references.

Lines 181-182: Provide some rationale for focusing on upd3.

Lines 208-209: Maybe reframe the setup of this section to be more direct, i.e., to provide support that impacts on metabolism are not the cause of the double mutant larval arrest. As written, the text leads the reader to expect to see assessment of many potential underlying causes.

Lines 219-231: It would be interesting to assess if JAK/STAT signaling is altered in other tissues when Ldh has been knocked down specifically in one tissue in the Gpdh1 mutant background (like was done for Figure 2).

Minor concerns

Line 130: missing the word “in” after “cells”

Error bars on all graphs: Since the color of the error bars is exactly the same as the symbol color, it is difficult to see the error bars. Perhaps change the color to black or a lighter shade of the symbol color and put the error bars on top.

Figure 3: At first glance, it isn’t clear that the factors listed are DEGs in the double mutant. To make this clearer from the figure alone, perhaps put the text on top of a green box.

Reviewer #3: In this study entitled “Glycolytic disruption restricts Drosophila melanogaster larval growth via the cytokine Upd3”, the authors describe a systemic signaling defect arising as a consequence of metabolic disruption in a specific tissue, primarily the muscle.

Loss of Gpdh1 and Ldh in the muscle leads to elevated Upd3 that drives developmental arrest by affecting the PG gland and dampened 20HE signaling. As a result, when the authors conduct Upd3 LOF in the Gpdh and Ldh double LOF condition they are able to restore the developmental arrest, thereby showing that the mediator of arrest is not a metabolic distress but rather a systemic signalling defect.

Overall, the study is interesting and implicates the importance of tissue specific metabolic state as a significant controller of organismal homeostasis.

I find the manuscript overall very important as it is in the realms of an area which is not thoroughly understood. The introduction is indeed very well written and positions the work very well. This raises the expectation when reading the manuscript, which however, then remains limited in terms of the core conceptual advancement that the study makes. Any understanding of how tissue specific metabolic states become deciding factors in coordinating organismal level development remains unanswered. As also pointed by the authors in the introduction, where this area of research is a relatively new, I suggest the authors to elaborate on this context as it is needed to strengthen the core problem posed in the study. Specifically:

1. Conceptual comments:

a. What is the relevance of the glycolytic state of the muscle with respect to the rest of the animal in homeostatic condition? What conditions influence this metabolic state is not described.

b. What is the lactate produced or G3P produced in normal homeostasis doing? Are they sensed by the PG to keep STAT activity in check allowing 20HE production and developmental continuity?

c. Does lactate supplementation in Ldh, Gpdh double mutant restore the growth delay? I understand the signaling downstream that the authors provide the data for when the tissue undergoes a disruption of metabolic homeostasis but in homeostasis, any understanding of the metabolites and what they contribute to remains unaddressed.

d. Finally, any physiological state that correlates with Ldh/gpdh lof condition is not mentioned. This will help the study and the readers with a context and a perspective that is currently missing. I strongly feel a physiological state recapitulating the Ldh/gpdh lof condition will tremendously help the current narrative.

2. Experimental comments:

a. have the authors checked for other Upds? Even if negative result this will be important to know the specificity.

b. Have the authors driven Mef2>Gpdh1 RNAi in the Ldh mutant, the converse experiment, and do they recapitulate the same result?

c. How does the disruption in glycolytic state lead to JNK activation?

d. Does driving JNK or Upd3 activation in the muscle, lead to the same phenotype as ldh/gpdh1 LOF?

e. A model will be good in the end that summarizes the story.

Overall, it is a relevant piece of work and I recommend the authors to explore the physiological inputs, as it will broaden the current scope of the work tremendously.

**Have all data underlying the figures and results presented in the manuscript been provided?**

Reviewer #1: Yes

Reviewer #2: Yes

Reviewer #3: Yes

PLOS authors have the option to publish the peer review history of their article (what does this mean? ). If published, this will include your full peer review and any attached files.

**Do you want your identity to be public for this peer review?** For information about this choice, including consent withdrawal, please see our Privacy Policy .

Reviewer #1: **Yes: ** Jaakko Mattila

Reviewer #2: No

Reviewer #3: **Yes: ** Tina Mukherjee

**Figure resubmission:**
---

## [Decision Letter · Decision Letter 1]

15 Apr 2025

Dear Dr Tennessen,

We are pleased to inform you that your manuscript entitled "Glycolytic disruption restricts Drosophila melanogaster larval growth via the cytokine Upd3" has been editorially accepted for publication in PLOS Genetics. Congratulations!

Yours sincerely,

Ville Hietakangas

Academic Editor

PLOS Genetics

Pablo Wappner

Section Editor

PLOS Genetics

Aimée Dudley

Editor-in-Chief

PLOS Genetics

Anne Goriely

Editor-in-Chief

PLOS Genetics

Comments from the reviewers (if applicable):

Reviewer's Responses to Questions

**Comments to the Authors:**

Reviewer #1: The authors have addressed all of my concerns.

Congrats to the authors for their nice work.

Reviewer #2: The authors have satisfactorily addressed all of my major concerns and nearly all of the minor concerns. One minor issue that still needs to be fixed is making sure that all error bars or on top of the data points. This was not adjusted in any of the figures for the revision.

Reviewer #3: The authors have now sufficiently revised the draft. I have no further comments or pending concerns.

**Have all data underlying the figures and results presented in the manuscript been provided?**

Reviewer #1: None

Reviewer #2: Yes

Reviewer #3: Yes

PLOS authors have the option to publish the peer review history of their article (what does this mean? ). If published, this will include your full peer review and any attached files.

**Do you want your identity to be public for this peer review?** For information about this choice, including consent withdrawal, please see our Privacy Policy .

Reviewer #1: **Yes: ** Jaakko Mattila

Reviewer #2: No

Reviewer #3: **Yes: ** Tina Mukherjee

**Data Deposition**

http://datadryad.org/submit?journalID=pgenetics&manu=PGENETICS-D-24-01432R1

**Press Queries**

---

## [Editor Report · Acceptance letter]

PGENETICS-D-24-01432R1

Glycolytic disruption restricts Drosophila melanogaster larval growth via the cytokine Upd3

Dear Dr Tennessen,

We are pleased to inform you that your manuscript entitled "Glycolytic disruption restricts Drosophila melanogaster larval growth via the cytokine Upd3" has been formally accepted for publication in PLOS Genetics! Your manuscript is now with our production department and you will be notified of the publication date in due course.

With kind regards,

Anita Estes

PLOS Genetics

On behalf of:
